# Reduced serial dependence suggests deficits in synaptic potentiation in anti-NMDAR encephalitis and schizophrenia

Heike Stein [1,7], Joao Barbosa [1,7], Mireia Rosa-Justicia [1,2], Laia Prades[1], Alba Morató[1], Adrià Galan-Gadea[1], Helena Ariño[1], Eugenia Martinez-Hernandez [1,3], Josefina Castro-Fornieles [1,2,4], Josep Dalmau [1,3,4,5,6] & Albert Compte [1✉]

A mechanistic understanding of core cognitive processes, such as working memory, is crucial to addressing psychiatric symptoms in brain disorders. We propose a combined psychophysical and biophysical account of two symptomatologically related diseases, both linked to hypofunctional NMDARs: schizophrenia and autoimmune anti-NMDAR encephalitis. We first quantified shared working memory alterations in a delayed-response task. In both patient groups, we report a markedly reduced influence of previous stimuli on working memory contents, despite preserved memory precision. We then simulated this finding with NMDAR-dependent synaptic alterations in a microcircuit model of prefrontal cortex. Changes in cortical excitation destabilized within-trial memory maintenance and could not account for disrupted serial dependence in working memory. Rather, a quantitative fit between data and simulations supports alterations of an NMDAR-dependent memory mechanism operating on longer timescales, such as short-term potentiation.

[1] Institut d'Investigacions Biomèdiques August Pi i Sunyer (IDIBAPS), Carrer Rosselló 149, 08036 Barcelona, Spain. [2] Department of Child and Adolescent Psychiatry and Psychology, 2017SGR881, CIBERSAM, Institute Clinic of Neurosciences, Hospital Clínic, Carrer Villarroel 170, 08036 Barcelona, Spain. [3] Service of Neurology, Hospital Clínic, Carrer Villarroel 170, 08036 Barcelona, Spain. [4] Department of Medicine, University of Barcelona, Carrer Casanova 143, 08036 Barcelona, Spain. [5] Institució Catalana de Recerca i Estudis Avançats (ICREA)-IDIBAPS, Carrer Casanova 143, 08036 Barcelona, Spain. [6] Department of Neurology, University of Pennsylvania, 3400 Spruce St, Philadelphia, PA 19104, USA. [7] These authors contributed equally: Stein Heike, Barbosa Joao. ✉email: acompte@clinic.cat

The NMDA receptor (NMDAR) subserves memory mechanisms at several timescales, including sustained working memory delay activity[1,2] and different temporal components of synaptic potentiation[3–5]. In addition, hypofunction of NMDARs is linked to psychiatric disease, in particular schizophrenia[6], and it possibly contributes to abnormal working memory function in patients with schizophrenia[7,8]. Indeed, reduced prefrontal NMDAR density characterizes this disease[9]. Yet, the specific neural alterations by which NMDAR hypofunction could lead to memory deficits in schizophrenia are still under debate[7,8]. Here, we studied working memory function in healthy controls, patients with schizophrenia, and patients recovering from anti-NMDAR encephalitis (see "Methods" section and Supplementary Table 1). Anti-NMDAR encephalitis is characterized by an antibody-mediated reduction of NMDARs[10], accompanied by initial psychosis and long-lasting memory deficits[11,12]. The prevalence of positive symptoms during the early stages of the disease causes frequent misdiagnosis as a schizophrenia spectrum disorder[13,14]. Here, we tested patients that had overcome acute stages, and had progressed to a more stabilized period with some positive symptoms but dominated by negative and cognitive symptoms, comparable to those in stabilized schizophrenia patients[15]. Due to the parallels in neurobiology, clinical aspects, and cognition of the two diseases, we expected working memory deficits in anti-NMDAR encephalitis to qualitatively resemble those in schizophrenia. This correspondence allows linking alterations in working memory to the NMDAR in both patient groups.

We assessed memory alterations in a visuospatial delayed-response task (Fig. 1a) on two coexisting temporal scales: single-trial working memory precision as a proxy of active memory maintenance during short delays, and serial dependence of responses on previously memorized stimuli[16,17] (serial biases, Fig. 1b) as a read-out of passive information maintenance across trials. Our results show reduced serial dependence but intact working memory precision in both patient populations. Neural correlates of this task have been identified in monkey prefrontal cortex[18–20], inspiring computational models that can capture key aspects of neural dynamics and behavior[18,21,22]. The biophysical detail of these models permits to investigate how NMDAR hypofunction at different synaptic sites affects circuit dynamics and working memory. Candidate mechanisms are a disturbed balance between cortical excitation and inhibition (excitation/inhibition balance), as it is observed in schizophrenia and in studies using NMDAR antagonists (e.g., ketamine)[2,6,23,24], and alterations in NMDAR-regulated short-term synaptic potentiation[3–5,25]. In the modeling section of this study, we systematically test the potential of these candidate mechanisms for explaining our behavioral findings. We conclude that a reduction in short-term potentiation in a network model of working memory most parsimoniously reproduces the experimentally observed memory alterations in schizophrenia and anti-NMDAR encephalitis.

## Results

### Unaltered working memory precision in both patients groups.
First, we sought to identify alterations in single-trial working memory precision, as an indication of a possible dysfunction of activity-based memory maintenance. Meta-analyses report mainly negative findings for delay-dependent precision impairments in schizophrenia and ketamine studies[7,26] (but see ref. [27]). We calculated the circular standard deviation of bias-corrected response errors ("Methods") as an inverse estimate of precision for each participant and delay. Correcting for biases as a systematic source of error allowed us to estimate memory precision independently of serial biases. For all groups, precision decreased

equally with delay (Fig. 1c), indicating spared active working memory maintenance over short delays of up to 3 s in encephalitis and schizophrenia.

### Patients' memories are less biased towards previous memories.
Next, we tested whether NMDAR-related memory alterations could be observed at intermediate timescales by measuring serial dependence. Serial dependence is defined as a systematic shift of responses towards previously remembered, uncorrelated stimuli[16] (Fig. 1b), revealing that traces of recently processed stimuli persist in memory circuits and are integrated with new memories. Importantly, these attractive biases emerge over the trial's memory delay, indicating a dependence on memory processes[28,29]. In conditions without memory requirements, only small repulsive biases are present, possibly generated during perceptual processing[28–30]. To assess NMDAR-related differences in serial dependence, we modeled single-trial errors $\theta^e$ as a linear mixed model of delay length, group, and a non-linear basis function of the distance $\theta^d$ between consecutive stimuli[16,29] (derivative-of-Gaussian, DoG($\theta^d$), "Methods", Eq. (1); Supplementary Fig. 1), and we assessed the significance of fixed effects through ANOVA tables ("Methods").

Serial dependence explained only a small fraction of single-trial errors in working memory (conditional $R^2 = 0.03$ for the linear model presented in Eq. (1)), reflecting its small magnitude compared to the typical extent of response inaccuracies (Fig. 1c), but it depended strongly on relevant task factors: In accordance with previous results[28,29], we found a dependence of attractive bias strength on memory delay (delay × DoG($\theta^d$), (F(2,58) = 13.89, $p = 1e-5$). Moreover, biases differed between groups of participants (group × DoG($\theta^d$), F(2,49) = 9.68, $p = 0.0003$), especially when comparing groups for different delay lengths (group × delay × DoG($\theta^d$), F(4,58) = 8.45, $p = 2e-5$). Figure 1d–f shows linear model fits and average bias curves for 0, 1, and 3 s delays (see Supplementary Figs. 2–4 for single-subject bias curves and fits). Groupwise linear models (Eq. (2)) allowed to assess the delay dependence of biases within each population (delay × DoG ($\theta^d$)): For healthy controls, initially repulsive biases became gradually more attractive with delay length (F(2,17) = 26.91, $p = 6e-6$; Supplementary Fig. 5). Encephalitis patients showed a qualitatively similar, but reduced pattern (F(2,23) = 5.06, $p = 0.015$). In contrast, no attractive bias emerged over delay in patients with schizophrenia (F(2,16) = 1.31, $p = 0.30$). Rather, a repulsive bias dominated all delay lengths in this group (DoG($\theta^d$), F(1,16) = 9.07, $p = 0.008$). Post-hoc tests and between-group comparisons are reported in Fig. 1g–i.

Serial dependence is known to fade with increasing inter-trial intervals (ITI)[29]. We controlled for ITI length by including ITI × DoG($\theta^d$) as a covariate in our linear model ("Methods", Eq. (4); Supplementary Fig. 6): For each additional second of ITI, serial bias decreased by $0.46 \pm 0.12°$ (mean ± s.d.). However, group differences in serial dependence remained unchanged after including the covariate. The timescale of serial dependence was further defined by how many past trials influenced the current response. We observed a much weaker delay-dependent bias towards the penultimate trial, but there was no consistent evidence for group differences (Supplementary Fig. 7a–c).

### Antipsychotic medication does not explain group differences.
We also controlled for potential effects of antipsychotic medication in chlorpromazine equivalents (CPZ, "Methods") in light of significant group differences in CPZ estimates (Supplementary Table 1), and an association of CPZ with individual serial bias strength within groups (Supplementary Fig. 8). When including CPZ as a covariate ("Methods", Eq. (5)), delay-dependent biases

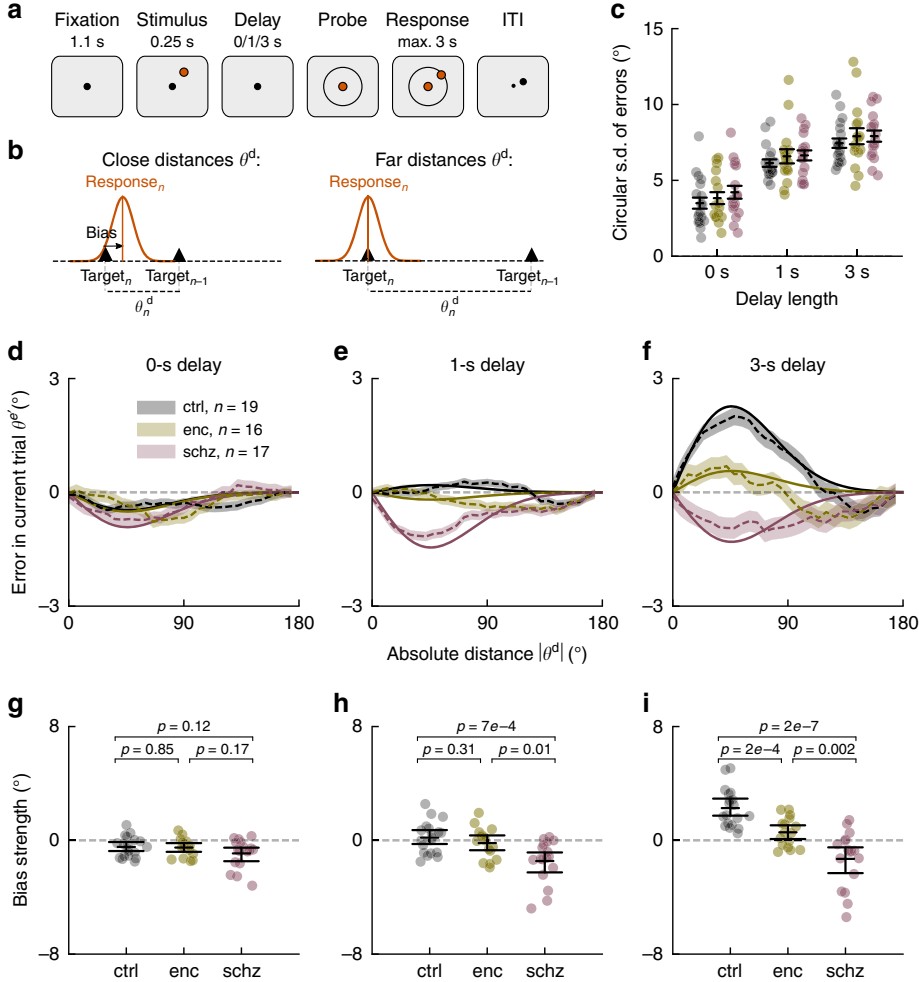

**Fig. 1 Reduced working memory-dependent serial dependence in anti-NMDAR encephalitis and schizophrenia. a** In each trial, subjects were to remember a stimulus that appeared for 0.25 s at a randomly chosen circular location with fixed distance from the center. Delay lengths varied randomly between trials (0, 1 or 3 s). Subjects made a mouse click to report the remembered location and started the next trial by moving the mouse back to the screen's center during the inter-trial-interval (ITI). **b** Serial dependence is measured as a systematic shift of responses towards previous target locations. Attractive effects depend on the distance $\theta^d$ between previous and current stimulus. **c** Precision for each subject and delay was inversely estimated as the circular s.d. of bias-corrected error distributions ("Methods"). For longer delays, participants' responses were less precise (delay, F(2,147) = 76.87, $p <$ 1e −16). There were no overall or delay-dependent group differences in precision (group, F(2,147) = 1.74, $p = 0.18$; group × delay, F(4,147) = 0.07, $p = .99$, all $p$-values from ANOVA). Error bars indicate ±s.e.m. **d–f**, Serial dependence by group and delay length. Serial dependence is calculated as the 'folded' error $\theta^{e'}$ for different $\theta^d$ (dashed lines; "Methods"). Solid lines show linear model fits ("Methods"), omitting intercepts and negative values of $\theta^d$. Shading, ±s.e.m. across pooled trials from $n = 19$ healthy controls (ctrl), $n = 17$ patients with schizophrenia (schz), and $n = 16$ patients with anti-NMDAR encephalitis (enc). **g–i** Individual (random coefficients; dots) and group estimates of serial bias strength (fixed effects; error bars indicate mean and bootstrapped 95% C.I. of the mean) by delay. **g** Serial dependence was repulsive in 0 s trials (DoG($\theta^d$), F(1,52) = 12.67, $p = 0.0008$), independently of group (group × DoG($\theta^d$), F(2,52) = 0.46, $p = 0.63$). **h** For 1 s trials, group differences in serial dependence emerged (group × DoG($\theta^d$), F(2,48) = 6.52, $p = 0.003$) between ctrl and schz ($t = 3.73$, $p = 7e-4$, Cohen's $d = 1.28$) and enc and schz ($t = 2.73$, $p = 0.01$, Cohen's $d = 0.98$). **i** After 3 s delay, both patient groups showed reduced biases compared to ctrl (group × DoG($\theta^d$), F(2,50) = 15.35, $p = 6e-5$; ctrl vs enc, $t = 4.14$, $p = 2e-4$, Cohen's $d = 1.45$; ctrl vs schz, $t = 6.44$, $p = 2e-7$, Cohen's $d = 2.21$, and enc vs schz, $t = 3.40$, $p = 0.002$, Cohen's $d = 1.22$). All $t$-tests, two-sided. In all panels, single data points show data from $n = 19$ healthy controls (ctrl), $n = 17$ patients with schizophrenia (schz), and $n = 16$ patients with anti-NMDAR encephalitis (enc).

still markedly differed between groups (Supplementary Fig. 8, caption). We designed two additional analyses to demonstrate the independence of group differences from the effect of antipsychotic medication: First, we showed that the difference in serial dependence persisted when we compared healthy controls to the unmedicated subset of encephalitis patients ($n = 12$ out of 16 encephalitis patients, Supplementary Fig. 9a–f). Second, we designed an analysis to test conservatively the group effect once we removed all the explanatory power of CPZ: We first

fitted single-trial errors $\theta^e$ as a function of CPZ and its one- and two-way interactions with delay and DoG($\theta^d$) in all subjects. On average, CPZ in patients with schizophrenia (370.6 ± 462.4 mg day$^{-1}$, mean ± s.d.) explained a reduction of 1.06° in biases in the 3 s delay condition, and only a reduction of 0.08° in encephalitis patients (with CPZ equivalents of 26.6 ± 52.7 mg day$^{-1}$, mean ± s.d.). Residuals of the linear model, now free of linear and multiplicative effects of CPZ estimates, were fitted as a function of group, delay, DoG($\theta^d$), and their interactions. Supplementary

Fig. 9g–l shows that group differences in memory-dependent biases remained marked (a reduction of 2.51° for schz, and 1.62° for enc in the 3 s delay condition) and highly significant even after conservatively controlling for CPZ.

**Encephalitis patients' biases increase with recovery.** We did not find correlations between individuals' bias estimates for 3 s delay trials and the severity of psychiatric symptoms for encephalitis or schizophrenia patients (Supplementary Fig. 8 and Supplementary Table 1). These between-subjects analyses were possibly underpowered, so we designed a within-subject longitudinal assessment for $n = 14$ encephalitis patients that returned for a follow-up session after 3–12 months (mean 8.5 months). As expected, clinical symptoms improved in these patients (Supplementary Table 2) and we found that serial dependence normalized with the patients' recovery (Eq. (8); Supplementary Fig. 10). Interestingly, for this subsample of encephalitis patients, positive and general symptoms measured in the PANSS scale correlated with serial dependence in the follow-up session (PANSS pos, $r = -0.70$, C.I. $= [-0.90, -0.26]$, $p = 0.006$; PANSS gen, $r = -0.62$, C.I. $= [-0.87, -0.13]$, $p = 0.02$), but again not significantly in the baseline session (PANSS pos, $r = -0.38$, C.I. $= [-0.76, 0.19]$, $p = 0.19$; PANSS gen, $r = -0.02$, C.I. $= [-0.54, 0.52]$, $p = 0.94$), although the direction of the effect was congruent between the two sessions. Moreover, patients with a stronger longitudinal normalization of biases improved more on the scale of positive symptoms (PANSS pos) in the follow-up session, when compared to the baseline session, $r = -0.54$, C.I. $= [-0.83, -0.02]$, $p = 0.04$ (Supplementary Fig. 10g; all correlations, Pearson's $r$).

Together, our experimental results show no differences in single-trial memory maintenance, but a strong reduction of delay-dependent biases in anti-NMDAR encephalitis that ameliorates with patients' recovery, and a complete absence of attractive biases in patients with schizophrenia. These findings are not explained by ITI length, general response correlations between trials (Supplementary Fig. 7d–f), response biases with respect to cardinal directions (Supplementary Fig. 11), or medication (Supplementary Fig. 8). Our conclusion is thus that alterations at the neural circuit level, related to NMDAR hypofunction, reduce serial dependence gradually, up to the point of completely disrupting attraction to previous stimuli. A prevailing idea associates NMDAR hypofunction in schizophrenia primarily to synapses onto GABAergic interneurons[23], while the role of NMDARs in working memory has been emphasized in synapses between pyramidal neurons[1,2,21]. Alternatively, NMDARs could be involved in mechanisms directly associated with the generation of serial biases, such as short-term plasticity[18,22,31]. To assess these mechanistic explanations comparatively, we simulated consecutive trials of a spatial working memory task in a spiking neural network model of the prefrontal cortex[21] (Fig. 2a). Prefrontal cortex not only holds working memory contents in an activity-based code[19,20], but also keeps long-lasting latent (possibly synaptic) memory traces that produce serial dependence[18].

**NMDAR hypofunction in a prefrontal working memory circuit.** We modeled a local prefrontal circuit, composed of neurons selective to the locations presented in the spatial working memory task. We used a network of excitatory and inhibitory neurons recurrently connected through AMPAR-, NMDAR- and $GABA_AR$-mediated synaptic transmission in which persistent delay firing emerges from attractor dynamics (Fig. 2a, Supplementary Fig. 12; "Methods"). As proposed by the previous studies[18,22,31], we modeled serial dependence as an effect of short-

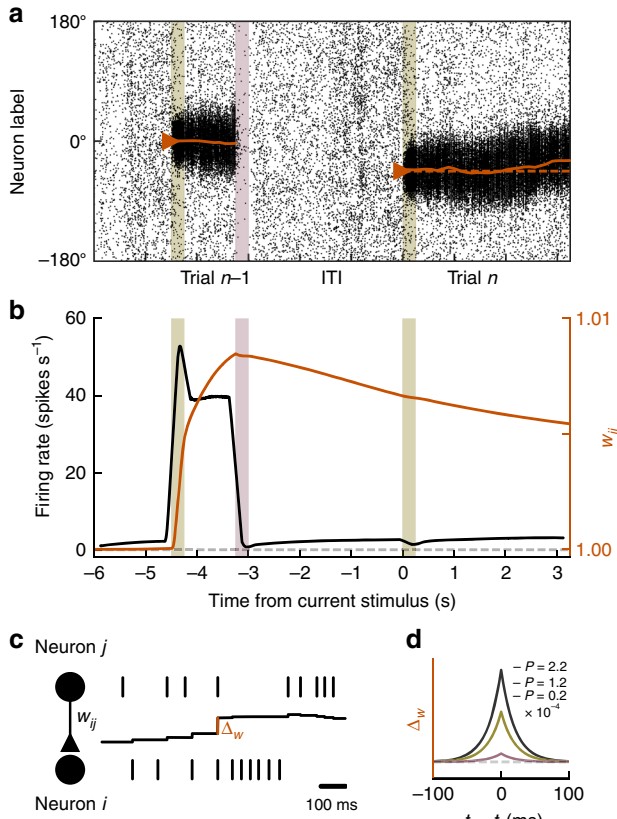

**Fig. 2 Ring attractor network with synaptic STP shows serial dependence.** Simulations of two consecutive working memory trials (current trial $n$, previous trial $n-1$) in a spiking neural network model with bump-attractor dynamics ("Methods"). **a** Spike times ($x$-axis) of excitatory neurons, ordered on $y$-axis by preferred angular location. Colored bars in **a**, **b** mark previous and current stimulus onset times (olive) and previous response (red). The solid orange line shows the population vector decoded from firing rates (sliding windows of 250 ms). In trial $n$, the active memory representation got biased towards the memory representation in trial $n-1$. **b** Firing rate (black) and potentiated weight trace $w_{ij}$ for neurons at 0° (orange) averaged over 1,000 trials and 20 neurons centered around 0°. Spiking activity and synaptic strength increased during trial $n-1$ delay and decreased after the response. At current stimulus onset, information about trial $n-1$ remained only in the potentiated weight trace. To facilitate interpretation, we excluded trials for which any neuron participated in previous and current-trial delay activity (i.e., showed firing rates >10 spikes $s^{-1}$ after stimulus onset in trial $n$). **c, d** Associativity and decay of modeled STP. The strength of each individual synapse is determined by $w_{ij}$ (**c**, middle black trace), which is potentiated at each spike by an amount $\Delta_w$ that depends on the relative spike times $t_j$ and $t_i$ of pre- and postsynaptic neurons, respectively, and on the potentiation factor $P$ that is chosen to represent different strengths of STP (different colored lines in (**d**); "Methods", Eqs. (15) and (16)), and it is reduced by an amount relative to the synaptic strength at each presynaptic spike, resulting in activity-dependent decay (Eq. (17)).

term plasticity that builds up at delay-active recurrent excitatory synapses and maintains information during the ITI in a sub-threshold stimulus representation not reflected in firing rate selectivity (Fig. 2b, "Methods"). We implemented an associative mechanism of short-term potentiation (STP) that is NMDAR-dependent and upregulates glutamatergic efficacy, consistent with a long-lasting increase in the probability of presynaptic neurotransmitter release[3,4]. As described in refs. [3,4], this efficacy

increase undergoes activity-dependent decay (Fig. 2c). In our simulations, stimulus-specific potentiated synaptic traces persisted through the ITI and attracted the next trial's memory representation progressively over the course of the delay[22,31]. To mimic memory-independent repulsive biases[29,30], current stimulus inputs were slightly shifted away from previous stimulus values by a fixed value[31] ("Methods"). This shift represents adaptation effects in sensory regions and is therefore not affected by local circuit alterations in prefrontal cortex.

We assessed the effects of NMDAR dysfunction on serial dependence at three potential synaptic sites: based on the reported NMDAR-dependence of STP[3–5], NMDAR hypofunction would reduce the strength of STP at excitatory synapses and disrupt delay-dependent biases (hypothesis I: reduced STP). Also, we tested the explanatory potential of reduced NMDAR-mediated synaptic transmission. In particular, we tested cortical disinhibition[27], caused by diminished NMDAR efficacy at inhibitory interneurons (hypothesis II: reduced $g_{EI}$), and the hypofunction of NMDARs at recurrent excitatory synapses, leading to diminished delay activity[2,32,33] (hypothesis III: reduced $g_{EE}$). To assess each of these mechanisms, we independently varied STP strength, $g_{EI}$ and $g_{EE}$, and we read out "behavioral responses" after 0, 1, and 3 s from population activity in our network simulations ("Methods"). Then, we fitted a linear model to measure bias strength in each condition (Eq. (18), Supplementary Fig. 13). We sought to identify which mechanisms could independently reproduce the patterns of reduced and absent biases observed in patients, and their dependence on working memory delay (Fig. 1).

**Reduced STP but not E-I imbalance disrupts memory biases.** We found that both hypotheses I and III were qualitatively consistent with our experimental results: NMDAR hypofunction (whether reducing STP or $g_{EE}$) reduced the strength of serial dependence (Fig. 3a, c, orange). In contrast, hypothesis II was discarded by our simulations: reducing $g_{EI}$ increased serial dependence (Fig. 3b, orange), contrary to our experimental results, and quickly led to network disinhibition, causing previous-trial delay activity to spontaneously reemerge in the ITI (Supplementary Fig. 14). Both for reduced $g_{EI}$ and reduced $g_{EE}$, the percentage of outlier responses (where errors $|\theta^e| > 57.3°$, i.e. 1 radian) quickly rose as the network lost the stability of one of its two states (spontaneous activity for reduced $g_{EI}$, and persistent delay activity for reduced $g_{EE}$, dashed vertical lines in Fig. 3b, c), as illustrated in Supplementary Fig. 14. Moreover, we noted that memory precision was slightly affected by all three manipulations (Fig. 3a–c), in contrast with our behavioral findings (Fig. 1b), but consistent with other studies with longer delays[27]. Delay length and task complexity could be important factors to detect NMDAR-related differences in memory precision.

In addition, we found that hypotheses I and III could be disambiguated based on biases produced by the different linear models in 0, 1, and 3 s delays (Fig. 3d–f). Even for the lowest value of $g_{EE}$ within the stable network regime (Fig. 3c), attractive biases increased with delay (Fig. 3f). While this manipulation can qualitatively reproduce decreased delay-dependent biases in the encephalitis group, it is incompatible with our results for patients with schizophrenia (Fig. 1), who do not develop attractive biases in memory trials. In contrast, reduced STP at recurrent excitatory synapses captured a pattern of equally strong repulsive biases for all delay lengths (Fig. 3d). Note that these findings also hold for a network with STP (and NMDAR-dependent reductions in STP) in inhibitory interneurons[34] (Supplementary Fig. 15). Based on our simulations, we conclude that the disruption of STP, a mechanism operating on a longer timescale than activity-based

memory maintenance, provides a plausible explanation for altered serial dependence as observed in schizophrenia and anti-NMDAR encephalitis.

## Discussion

In this study, we assessed working memory alterations in two patient groups linked to NMDAR hypofunction, and hypothesized that their shared clinical and neurobiological features should be reflected in qualitatively similar behavioral patterns. In accordance with this reasoning, we found a drastic reduction of working memory serial dependence both in patients with anti-NMDAR encephalitis and schizophrenia, as compared to healthy controls. In contrast, we did not find memory maintenance deficits on timescales of a few seconds, suggesting that cognitive deficits in these patients[8,12] might be partly explained by the disruption of long-lasting, inactive memory traces, and a lacking integration of past and current memories. Our modeling results show that simple alterations in cortical excitation (hypotheses II and III), as proposed by current theories of NMDAR hypofunction in schizophrenia[6,24,27], cannot fully explain these behavioral findings. Instead, altered serial dependence is mechanistically accounted for by a disruption in slower dynamics, here specified as NMDAR-dependent associative STP (hypothesis I) that is triggered by sustained delay activity and influences memory representations in upcoming trials. Our results suggest that clinical reports of short-term memory alterations in schizophrenia and anti-NMDAR encephalitis could be understood in the light of reduced synaptic potentiation[25]. This is consistent with in vitro studies, which have demonstrated the dependence of STP on specific subunit components of the NMDAR[3,4], and reduced STP in genetic mouse models of schizophrenia[35]. Importantly, our modeling is not incompatible with altered cortical excitatory or inhibitory tone as a result of hypofunctional NMDARs. Rather, it states the necessity of assuming alterations in a mechanism operating on longer timescales, such as STP. For instance, diminished STP alongside symmetric effects on both E-E and E-I synapses could maintain the excitation/inhibition balance and thus stable delay activity, while interrupting passive between-trial information maintenance.

Future studies should address the effects of pharmacological NMDAR blockade on serial dependence. These studies could unequivocally confirm the role of the NMDAR for trial-history effects in working memory, and at the same time allow to ask more specific questions: On the one hand, serial dependence effects under different NMDAR antagonists should vary according to how blocking specific NMDAR subunits modulates synaptic potentiation at different timescales[3]. Our results cannot address subunit specificity because anti-NMDAR encephalitis (and possibly schizophrenia[9]) is associated with hypofunction of the GluN1 subunit, which is contained in all NMDARs[36,37]. On the other hand, pharmacological studies in combination with neural recordings could reveal how trial-history representations are affected by the blockade of NMDARs[18,38]. In rodents, long-term pharmacological experiments during behavior could be complemented with in vitro studies to assess STP directly. Finally, pharmacological studies would clarify if the alterations in serial dependence occur as a result of acute NMDAR hypofunction or whether they depend on compensatory changes in STP that arise after early, acute phases of cortical excitation/inhibition imbalance in these diseases (e.g., as a long-term adjustment of the probability of presynaptic neurotransmitter release).

We showed how working memory in the two investigated diseases is altered in a parallel way, and how these alterations are parsimoniously explained by manipulating a single, NMDAR-dependent synaptic variable in our model. However, substantial

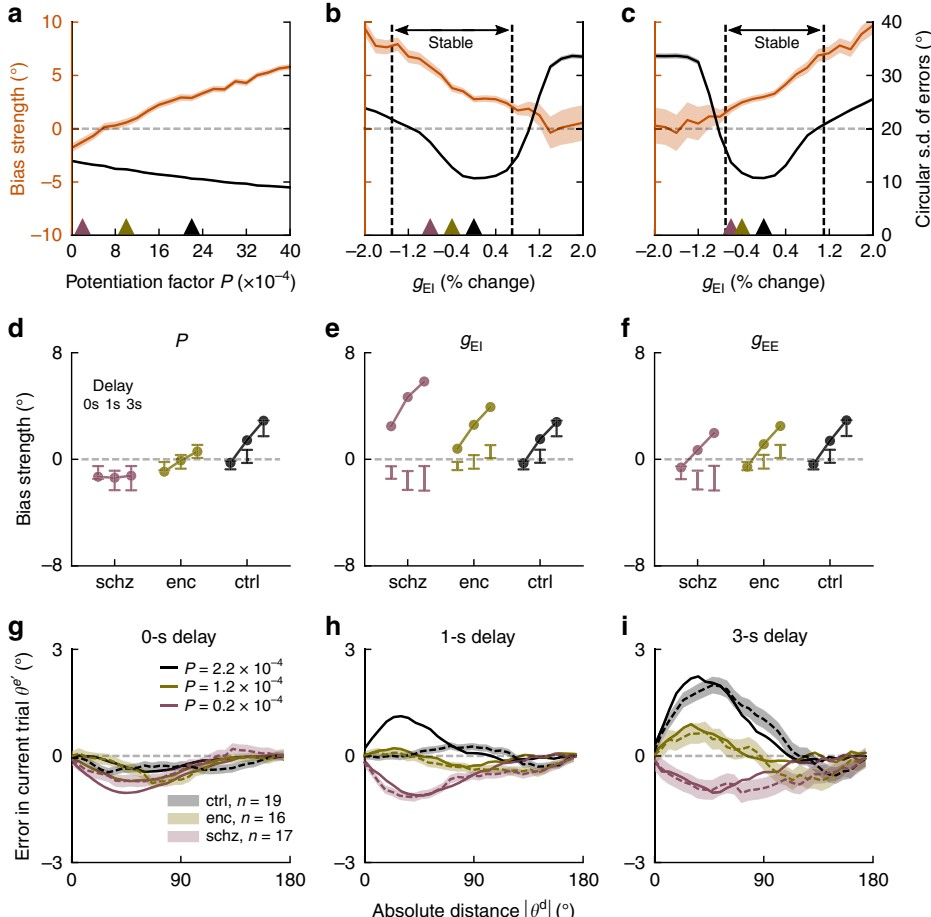

**Fig. 3 Altered STP simulates reduced serial dependence in spiking neural networks. a–c** Serial dependence (orange, bias coefficients from linear model, "Methods") and precision (black, circular s.d. of errors) as a function of model parameters in 3 s delay trials (20,000 trials per parameter value). Vertical dashed lines indicate transition to unstable network regimes for which more than 10% of trials were outliers ($|\theta^e| > 57.3°$, i.e., 1 radian). Shading, 95% C.I. for regression estimates of bias coefficients in simulated responses. **a** Serial dependence decreased gradually when decreasing STP (potentiation factor $P$), while the network remained stable for all simulated values of $P$. Precision changed slightly as a function of STP. **b** Cortical disinhibition via decreased $g_{EI}$ augmented serial bias while strongly affecting precision and stability, either due to instability of persistent activity (right, Supplementary Fig. 14b), or due to instability of spontaneous activity (left, Supplementary Fig. 14a). **c** Lowering recurrent cortical excitation ($g_{EE}$) led to the opposite pattern, decreasing biases. **d–f** Delay dependence of biases for each group, as defined by parameter values in (**a–c**), (respectively colored triangles). Points depict mean bias strength (over 20,000 trials) for each parameter value. For comparison, error bars indicate 95% CI for bias strength obtained from $n = 19$ healthy controls (ctrl), $n = 17$ patients with schizophrenia (schz), and $n = 16$ patients with anti-NMDAR encephalitis (enc) (reordered from Fig. 1g-i). **d** Lowering STP strength reproduced the experimental data. In **e, f** reduction of NMDAR conductances ($g_{EI}$ or $g_{EE}$) did not reproduce group and delay dependencies of experimental biases. **g–i** Solid lines, simulated serial dependence by delay length for different values of $P$, indicated by colored triangles in (**a**) (20,000 trials per potentiation level $P$). Dashed lines with error bars, serial dependence in encephalitis, schizophrenia, and healthy controls. Bias calculated as averaged 'folded' error $\theta^{e\prime}$ for binned absolute previous-current distances $\theta^d$. Shading, ±s.e.m. Compare to Supplementary Fig. 15 for a network with STP (and STP disruptions in patients) in both E–E and E–I connections.

neurobiological heterogeneity must underlie the differences in epidemiology and longitudinal development of schizophrenia and autoimmune anti-NMDAR encephalitis[39]. Under this reasoning, we cannot exclude that distinct biological mechanisms in our two patient groups might lead to convergent patterns of working memory processing. For instance, our modeling shows that encephalitis patients' biases could also be explained qualitatively by a reduced excitation-to-inhibition ratio in the memory circuit (Fig. 3f), consistent with task-related fMRI BOLD activity in ketamine[33], and the effect of NMDAR antagonists on single-cell firing rates in monkey PFC[2]. In contrast, we could not confirm the findings of previous modeling work of schizophrenia, postulating that deficits in working memory precision and higher susceptibility to distractors[40,41] or alterations in probabilistic reasoning[42] could be explained by an increased excitation-to-

inhibition ratio, leading to cortical disinhibition. This mechanistic alteration cannot replicate serial dependence deficits in schizophrenia in our model (Fig. 3b, e). Reduced short-term plasticity, in contrast, would predict reduced working memory precision after long memory delays (Fig. 3a, see also ref.[43]), and higher susceptibility to distractors[44] in line with reported behavior in schizophrenia[41], which was previously proposed to reflect an excessive excitation-to-inhibition ratio. In addition, some incongruences with previous findings might be explained by the acuteness of the patients' condition, with more acute or psychotic stages being connected with patterns of disinhibition, and less acute stages with residual alterations in synaptic plasticity, but not cortical excitation. Alternatively, mechanisms not considered in our model could be at play. For instance, NMDAR dysfunction could negatively affect long-range connectivity[45–47] between trial

history-tracking areas[38] and areas that hold current working memory contents (like prefrontal cortex), and in this way impede the integration of previous with current memories. Note, however, that recent combined experimental and theoretical work in primate and human prefrontal cortex shows how both past and current memories are jointly represented in prefrontal cortex, and how their interaction subserves serial dependence[18].

Our findings advance the conceptual understanding of working memory alterations in schizophrenia and anti-NMDAR encephalitis, as they demonstrate a selective disruption of information carryover between trials, reflected by a reduction of serial dependence that is robustly found in neurotypical subjects[17]. We found several indicators of clinical relevance for our finding. First, as anti-NMDAR encephalitis patients recovered, their biases normalized in the direction of healthy controls (Supplementary Fig. 10a–c). Second, the amount of this normalization correlated across patients with their improvement on a scale that measures positive symptoms (Supplementary Fig. 10g), indicating a potential relation between psychotic symptoms and reductions in serial dependence. Third, both the alterations in serial dependence and the strength of positive symptoms were higher for patients with schizophrenia than for the anti-NMDAR encephalitis group. Still, studies with larger sample sizes are needed to confirm the relation of psychotic symptoms and reduced serial biases at the subject-level, which in our study did not reach significance for two out of three analyses in patients with schizophrenia and anti-NMDAR encephalitis (Supplementary Fig. 8 and "Results").

Serial dependence could also reflect a clinically relevant dimension which is not or only mildly related to the assessed psychiatric scales. In this sense, it has been argued that serial dependence could facilitate information processing in temporally coherent real-world situations[17]. Alternatively, serial biases could be the mere by-product of long-lasting cellular or synaptic mechanisms that support memory stabilization during working memory delays[48]. Our study is in line with previous findings of reduced susceptibility to proactive interference in schizophrenia[49,50]. However, while proactive interference is mainly discussed in the context of cognitive control, the limited complexity of our task restricts possible interpretations of reduced between-trial interference and supports the role of reduced residual memory traces. Moreover, thanks to our task's well-studied single-neuron correlates[18–20] and biophysical models[18,19,21] and the comparison with anti-NMDAR encephalitis patients, we provide a specific mechanistic model of synaptic deficits leading to reduced previous-trial interference in schizophrenia.

Interestingly, a reduction in serial dependence has recently been reported for patients with autism[51], a disease also associated with NMDAR hypofunction[52] and alterations in synaptic potentiation[25]. Further, as for autism, our findings of reduced serial dependence are compatible with normative accounts of information processing in schizophrenia. Classic theories and recent studies have reported an underweighting of past context, or in Bayesian terms, learned priors, and an overweighting of incoming perceptual information in patients with schizophrenia[42,53,54] and NMDAR hypofunction[55]. Long-lived traces of past stimuli could serve as Bayesian priors to perception and memory, and a disruption of STP might be regarded as a biological implementation of a reduced usage of priors in schizophrenia and anti-NMDAR encephalitis.

## Methods

**Experimental sample.** We included $n = 16$ patients with anti-NMDAR encephalitis (enc), $n = 17$ patients with schizophrenia or schizoaffective disorder ($n = 12$ and $n = 5$, respectively; schz), and $n = 19$ neurologically and psychiatrically

healthy control participants (ctrl), all with normal or corrected vision. Behavioral data from $n = 14$ healthy controls has been included in a previous study[18]. Psychiatric diagnoses (or the absence thereof for controls) were confirmed using the Structured Clinical Interview for DSM IV (SCID-I)[56]. Patients diagnosed with anti-NMDAR encephalitis were recruited from different centers ($n = 14$ in Spain, $n = 1$ in Germany and $n = 1$ in the United Kingdom) at the moment of hospital discharge and completed the experiment around 5.5 months after disease onset (median, interquartile range i.q.r. = 3.7–7.2 months). All patients fulfilled clinical diagnostic criteria of anti-NMDAR encephalitis with confirmation of CSF IgG antibodies against the GluN1 subunit of the NMDAR[57]. All subjects were tested in our laboratory for antibodies against NMDAR in serum[36] and all healthy controls and patients with schizophrenia were seronegative. Anti-NMDAR encephalitis is known to have a prolonged process of recovery after the acute stage of the disease[58], and patients in the prolonged recovery phase still suffer from cognitive deficits as has been previously described in cohorts with long follow-up[12]. All patients were sufficiently recovered to participate in the testing procedure. Controls and patients with schizophrenia were recruited from the Barcelona area and from Hospital Clínic (Barcelona, Spain), respectively. Patients with schizophrenia were tested 35.0 months after diagnosis (median, i.q.r. = 16.0–69.5 months) and were clinically stable at the time of testing. All participants (and, in the case of minors of age, their legal guardians) provided written informed consent and were monetarily compensated for their time and travel expenses, as reviewed and approved by the Research Ethics Committee of Hospital Clínic. All subjects were assessed for psychiatric symptoms and functionality through a battery of standard tests including the Spanish versions of the Positive and Negative Syndrome Scale (PANSS)[59], the Young Mania Rating Scale (YMRS)[60], the Hamilton Depression Rating Scale (HAM-D)[61] and the Global Assessment of Functioning Scale (GAF)[62]. Finally, the dose of antipsychotic medication at the moment of testing was estimated as chlorpromazine equivalent (CPZ, mg day$^{-1}$)[63]. For a demographic and clinical overview of the populations, please refer to Supplementary Table 1.

**Experimental task protocol and behavioral testing.** Participants completed two 1.5 h sessions performing a visuospatial working memory task described in Fig. 1a. In each session, participants were asked to complete 12 blocks of 48 trials. However, some participants did not complete all blocks (on average, participants completed $1114.1 \pm 134.4$ trials (mean ± s.d., ctrl), $1086.0 \pm 189.9$ trials (enc), and $1030.6 \pm 192.8$ trials (schz)).

For stimulus presentation, we used Psychopy v3.1.5 on Python 2.7, running on a 17" HP ProBook laptop. Each trial began with the presentation of a central black fixation square on a gray background ($0.5 \times 0.5$ cm) for 1.1 s. A single colored circle (stimulus, diameter 1.4 cm, 1 out of 6 randomly chosen colors with equal luminance) was then presented during 0.25 s at one of 360 randomly chosen angular locations at a fixed radius of 4.5 cm from the center. The stimulus was followed by a randomly chosen delay of 0 (16.67% of trials), 1 (66.67% of trials), or 3 s (16.67% of trials) in which only the fixation dot remained visible (except for 0 s trials, where the stimulus remained visible until the participant started to move the cursor). When the fixation dot changed to the stimulus' color (probe), participants were asked to respond by making a mouse click at the remembered location (response). A white circle indicated the stimulus' radial distance, so participants only had to remember the angular position. After the response, the cursor had to be moved back to the fixation dot to start a new trial (ITI). Participants were instructed to maintain fixation during the fixation period, stimulus presentation, and memory delay and were free to move their eyes during response and when returning the cursor to the fixation dot.

**Error and serial dependence analysis.** Response errors $\theta_n^e$ in trial $n$ were measured as the angular distance between response and target. To exclude errors due to guessing or motor imprecision, we only analyzed responses within an angular distance of 1 radian and a radial distance of 2.25 cm from the stimulus. Further, we excluded trials in which the time of response initiation exceeded 3 s, and trials for which the time between the previous trial's response probe and the current trial's stimulus presentation exceeded 5 s. In total, $2.6 \pm 4.2\%$ (mean ± s.d., ctrl), $4.8 \pm 6.9\%$ (enc) and $7.5 \pm 9.6\%$ (schz) of trials per participant were rejected (but only $0.1 \pm 0.2\%$ (ctrl), $0.4 \pm 0.5\%$ (enc) and $0.6 \pm 0.7\%$ (schz) of trials were excluded due to angular response errors).

We then measured serial dependence as the error in the current trial as a function of the circular distance between the previous and the current trial's target location. Figure 1c–e depict 'folded' serial dependence: We multiplied trial-wise errors $\theta_n^e$ by the sign of the previous-current distance, $\theta_n^d$: $\theta_n^{e\prime} = \theta_n^e * \text{sign}(\theta_n^d)$, and then binned data based on absolute values $|\theta_n^d|$. Errors $\theta_n^{e\prime}$ were then averaged for each $|\theta_n^d|$ in sliding windows with size $\pi/3$ in steps of $\pi/30$. Positive mean folded errors should be interpreted as attraction towards the previous stimulus and negative mean folded errors as repulsion away from the previous location. In all figures including bias curves, s.e.m. are calculated across pooled trials from all subjects for each group and delay. For visualization, all values were transformed from radians to angular degrees.

**Linear (mixed) models.** We modeled signed errors $\theta_{nm}^e$ in trial $n$ and subject $m$ using linear mixed models that included the dummy-coded variables group (ctrl,

enc or schz) and delay (0, 1, or 3 s), and a nonlinear function of previous-current stimulus distance $\theta_{nm}^d$, $\mathrm{DoG}(\theta_{nm}^d)$, which has been used for modeling serial dependence[16,29]. $\mathrm{DoG}(\theta_{nm}^d)$ is the normalized first derivative of a Gaussian with fixed location hyperparameter $\mu = 0$. Its scale parameter $\sigma$ was determined using cross-validation as explained below (see also Supplementary Fig. 1). Our main linear model is:

$$
\begin{aligned}
\theta_{nm}^e = \beta_0 &+ \beta_{1,g}\mathrm{group}_{nm} + \beta_{2,d}\mathrm{delay}_{nm} - \beta_3\mathrm{DoG}(\theta_{nm}^d) \\
&+ \beta_{4,g,d}\mathrm{group}_{nm}\mathrm{delay}_{nm} - \beta_{5,g}\mathrm{group}_{nm}\mathrm{DoG}(\theta_{nm}^d) \\
&- \beta_{6,d}\mathrm{delay}_{nm}\mathrm{DoG}(\theta_{nm}^d) - \beta_{7,g,d}\mathrm{group}_{nm}\mathrm{delay}_{nm}\mathrm{DoG}(\theta_{nm}^d) \\
&+ \gamma_{0,m} - \gamma_{1,m}\mathrm{DoG}(\theta_{nm}^d) - \gamma_{2,m,d}\mathrm{delay}_{nm}\mathrm{DoG}(\theta_{nm}^d) + \varepsilon_{nm}
\end{aligned}
\tag{1}
$$

$\beta$ coefficients estimate fixed, and $\gamma$ coefficients random effects. Coefficient subscripts $g$ and $d$ denote that a separate coefficient was estimated for different values of dummy-coded variables group or delay, respectively, resulting in a total of 18 $\beta$ coefficients for Eq. (1). Coefficient subscript $m$ denotes that a separate coefficient was estimated for each subject. Bias strength for a certain condition can then be read out as the sum of coefficients of all terms containing $\mathrm{DoG}(\theta_{nm}^d)$ and the dependence of bias strength on other variables is assessed by evaluating the significance of interaction terms containing $\mathrm{DoG}(\theta_{nm}^d)$ and the relevant variable. To measure response precision, bias-corrected response errors were defined as linear model residuals $\varepsilon_{nm}$ from Eq. (1). For each subject and delay, inverse response precision was then measured as the circular s.d. of $\varepsilon_{nm}$.

Group- (Eq. (2), Supplementary Fig. 5) and delay-wise (Eq. (3), Fig. 1g–i) linear models were defined as:

$$
\begin{aligned}
\theta_{nm}^e = \beta_0 &+ \beta_{1,d}\mathrm{delay}_{nm} - \beta_2\mathrm{DoG}(\theta_{nm}^d) - \beta_{3,d}\mathrm{delay}_{nm}\mathrm{DoG}(\theta_{nm}^d) \\
&+ \gamma_{0,m} - \gamma_{1,m}\mathrm{DoG}(\theta_{nm}^d) - \gamma_{2,m,d}\mathrm{delay}_{nm}\mathrm{DoG}(\theta_{nm}^d) + \varepsilon_{nm}
\end{aligned}
\tag{2}
$$

$$
\begin{aligned}
\theta_{nm}^e = \beta_0 &+ \beta_{1,g}\mathrm{group}_{nm} - \beta_2\mathrm{DoG}(\theta_{nm}^d) - \beta_{3,g}\mathrm{group}_{nm}\mathrm{DoG}(\theta_{nm}^d) \\
&+ \gamma_{0,m} - \gamma_{1,m}\mathrm{DoG}(\theta_{nm}^d) + \varepsilon_{nm}
\end{aligned}
\tag{3}
$$

The effect of covariates ITI length (Eq. (4)) and CPZ equivalent (Eq. (5)) were assessed as:

$$
\begin{aligned}
\theta_{nm}^e = \beta_0 &+ \beta_{1,g}\mathrm{group}_{nm} + \beta_{2,d}\mathrm{delay}_{nm} - \beta_3\mathrm{DoG}(\theta_{nm}^d) \\
&+ \beta_{4,g,d}\mathrm{group}_{nm}\mathrm{delay}_{nm} - \beta_{5,g}\mathrm{group}_{nm}\mathrm{DoG}(\theta_{nm}^d) \\
&- \beta_{6,d}\mathrm{delay}_{nm}\mathrm{DoG}(\theta_{nm}^d) - \beta_{7,g,d}\mathrm{group}_{nm}\mathrm{delay}_{nm}\mathrm{DoG}(\theta_{nm}^d) \\
&- \beta_8\mathrm{ITI}_{nm}\mathrm{DoG}(\theta_{nm}^d) + \gamma_{0,m} - \gamma_{1,m}\mathrm{DoG}(\theta_{nm}^d) \\
&- \gamma_{2,m,d}\mathrm{delay}_{nm}\mathrm{DoG}(\theta_{nm}^d) + \varepsilon_{nm}
\end{aligned}
\tag{4}
$$

$$
\begin{aligned}
\theta_{nm}^e = \beta_0 &+ \beta_{1,g}\mathrm{group}_{nm} + \beta_{2,d}\mathrm{delay}_{nm} - \beta_3\mathrm{DoG}(\theta_{nm}^d) \\
&+ \beta_{4,g,d}\mathrm{group}_{nm}\mathrm{delay}_{nm} - \beta_{5,g}\mathrm{group}_{nm}\mathrm{DoG}(\theta_{nm}^d) \\
&- \beta_{6,d}\mathrm{delay}_{nm}\mathrm{DoG}(\theta_{nm}^d) - \beta_{7,g,d}\mathrm{group}_{nm}\mathrm{delay}_{nm}\mathrm{DoG}(\theta_{nm}^d) \\
&- \beta_{8,d}\mathrm{CPZ}_{nm}\mathrm{delay}_{nm}\mathrm{DoG}(\theta_{nm}^d) + \gamma_{0,m} - \gamma_{1,m}\mathrm{DoG}(\theta_{nm}^d) \\
&- \gamma_{2,m,d}\mathrm{delay}_{nm}\mathrm{DoG}(\theta_{nm}^d) + \varepsilon_{nm}
\end{aligned}
\tag{5}
$$

Further, a conservative estimate of group effects when controlling for CPZ equivalents was obtained by first regressing trialwise errors as CPZ-dependent effects excluding random effects to not absorb variance related to the experimental group that subjects belonged to (notice dropped $m$ subscripts):

$$
\begin{aligned}
\theta_n^e = \beta_0 &+ \beta_1\mathrm{CPZ}_n + \beta_{2,d}\mathrm{CPZ}_n\mathrm{delay}_n - \beta_3\mathrm{CPZ}_n\mathrm{DoG}(\theta_n^d) \\
&- \beta_{4,d}\mathrm{CPZ}_n\mathrm{delay}_n\mathrm{DoG}(\theta_n^d) + \varepsilon_n
\end{aligned}
\tag{6}
$$

and subsequently modeling residuals $\varepsilon_n$ as main and interaction effects of group, delay, and $\mathrm{DoG}(\theta_{nm}^d)$ as described in Eq. (1) (Supplementary Fig. 9g–l).

Biases towards stimuli in trial $n − 2$ were measured by including distances to the penultimate stimulus, $\theta_{nm}^{d\prime}$,

$$
\begin{aligned}
\theta_{nm}^e = \beta_0 &+ \beta_{1,g}\mathrm{group}_{nm} + \beta_{2,d}\mathrm{delay}_{nm} - \beta_3\mathrm{DoG}(\theta_{nm}^d) \\
&+ \beta_{4,g,d}\mathrm{group}_{nm}\mathrm{delay}_{nm} - \beta_{5,g}\mathrm{group}_{nm}\mathrm{DoG}(\theta_{nm}^d) \\
&- \beta_{6,d}\mathrm{delay}_{nm}\mathrm{DoG}(\theta_{nm}^d) - \beta_{7,g,d}\mathrm{group}_{nm}\mathrm{delay}_{nm}\mathrm{DoG}(\theta_{nm}^d) \\
&- \beta_8\mathrm{DoG}(\theta_{nm}^{d\prime}) - \beta_{9,g}\mathrm{group}_{nm}\mathrm{DoG}(\theta_{nm}^{d\prime}) - \beta_{10,d}\mathrm{delay}_{nm}\mathrm{DoG}(\theta_{nm}^{d\prime}) \\
&- \beta_{11,g,d}\mathrm{group}_{nm}\mathrm{delay}_{nm}\mathrm{DoG}(\theta_{nm}^{d\prime}) + \gamma_{0,m} - \gamma_{1,m}\mathrm{DoG}(\theta_{nm}^d) \\
&- \gamma_{2,m,d}\mathrm{delay}_{nm}\mathrm{DoG}(\theta_{nm}^d) + \varepsilon_{nm}
\end{aligned}
\tag{7}
$$

Baseline and follow-up sessions in encephalitis patients and controls were compared by:

$$
\begin{aligned}
\theta_n^e = \beta_0 &+ \beta_1\mathrm{session}_n + \beta_{2,g}\mathrm{group}_n + \beta_{3,d}\mathrm{delay}_n \\
&- \beta_4\mathrm{DoG}(\theta_n^d) + \beta_{5,g}\mathrm{session}_n\mathrm{group}_n + \beta_{6,d}\mathrm{session}_n\mathrm{delay}_n \\
&+ \beta_{7,g,d}\mathrm{group}_n\mathrm{delay}_n - \beta_8\mathrm{session}_n\mathrm{DoG}(\theta_n^d) \\
&- \beta_{9,g}\mathrm{group}_n\mathrm{DoG}(\theta_n^d) - \beta_{10,d}\mathrm{delay}_n\mathrm{DoG}(\theta_n^d) \\
&- \beta_{11,g}\mathrm{session}_n\mathrm{group}_n\mathrm{DoG}(\theta_n^d) - \beta_{12,d}\mathrm{session}_n\mathrm{delay}_n\mathrm{DoG}(\theta_n^d) \\
&- \beta_{13,g,d}\mathrm{group}_n\mathrm{delay}_n\mathrm{DoG}(\theta_n^d) - \beta_{14,g,d}\mathrm{session}_n\mathrm{group}_n\mathrm{delay}_n\mathrm{DoG}(\theta_n^d) + \varepsilon_n
\end{aligned}
\tag{8}
$$

where $\mathrm{session}_n$ takes values 0 or 1 (baseline vs. follow-up). In this model, we did not include random effects due to increased model complexity and resulting difficulties in model convergence. For extended linear models in Eqs. (4), (5), (7), and (8), we compared nested models via Wald Tests to determine the optimal model complexity. Data was analyzed in Python 3.7. We used different packages from R statistics (version 3.6.3) through the 'rpy2' interface[64]. All linear mixed models were fitted, compared and statistically tested with packages 'lme4'[65] and 'lmerTest'[66], which calculates ANOVA tables for the fixed effects of the linear mixed model by estimating degrees of freedom and F values using Satterthwaite's method. For optimization, we used the 'optimx' package[67] 'nlimb' algorithm with a convergence tolerance of 0.003 and checked the consistency of parameter estimates with other optimization algorithms ('L-BFGS-B', 'bobyqa'). Note that the normality assumption of residuals was not met (normality test, $s^2 + k^2 = 4248.72$, $p < 1e{-}16$), but with only slightly diverting kurtosis (Fisher) = 3.37 and skewness = 0.12 parameters. Due to the large number of trials ($n = 52,394$), this should not compromise statistical inference[68]. Moreover, all effects of relevant task variables are visualized both in a model-based and model-free way to confirm their congruence.

**Basis function selection and hyperparameter cross-validation.** To determine the hyperparameter $\sigma$ used in Eqs. (1)–(8), we fitted errors $\theta_n^e$ in trial $n$ as a linear model including factors group, delay, and $\mathrm{DoG}(\theta_n^d)$ as described in Eq. (1), but excluding random effects:

$$
\begin{aligned}
\theta_n^e = \beta_0 &+ \beta_{1,g}\mathrm{group}_n + \beta_{2,d}\mathrm{delay}_n - \beta_3\mathrm{DoG}(\theta_n^d) \\
&+ \beta_{4,g,d}\mathrm{group}_n\mathrm{delay}_n - \beta_{5,g}\mathrm{group}_n\mathrm{DoG}(\theta_n^d) \\
&- \beta_{6,d}\mathrm{delay}_n\mathrm{DoG}(\theta_n^d) - \beta_{7,g,d}\mathrm{group}_n\mathrm{delay}_n\mathrm{DoG}(\theta_n^d) + \varepsilon_n
\end{aligned}
\tag{9}
$$

while setting Gaussian hyperparameters $\mu = 0$ and $\sigma \in [0.2, 1.8]$ (in radians). For each value of the scale parameter $\sigma$, we used a stratified cross-validation procedure, fitting the linear model to 67% of the trials from each subject and testing the prediction in the left-out 33% of trials. Performance for each $\sigma$ was evaluated using the mean squared error (MSE) of predictions from 1000 cross-validation repetitions. $\sigma$ was chosen so as to minimize the MSE obtained by the linear model, yielding $\sigma = 0.8$ (Supplementary Fig. 1).

To test whether a linear model with repulsive biases at high distances $|\theta_n^d|$ fitted our data more parsimoniously, we compared cross-validation MSE for linear models with first- and third-derivative-of-Gaussian basis functions (Supplementary Fig. 1). We repeated the hyperparameter fitting procedure described above for the third-derivative-of-Gaussian model using hyperparameters $\mu = 0$ and $\sigma \in [0.6, 2.0]$ rad. As the first-derivative-of-Gaussian model produced smaller MSE in the cross-validation procedure, we discarded the third-derivative-of-Gaussian model. Thus, all linear model results reported in this manuscript correspond to the first-derivative-of-Gaussian model.

**Confidence intervals and effect sizes.** We compared single-subject bias estimates between groups using post hoc $t$-tests. Effect sizes for these comparisons were estimated as Cohen's d, defined as $d = \frac{\mu_1 - \mu_2}{s}$ for independent samples, where $s$ is the pooled standard deviation: $s = \sqrt{\frac{(n_1 - 1)s_1^2 + (n_2 - 1)s_2^2}{n_1 + n_2 - 2}}$, and as $d = \frac{t}{\sqrt{n}}$ for related samples. For correlations of individual subjects' biases with symptoms, we used Pearson correlation and calculated parametric 95% confidence intervals ('CIr' function from the 'psychometric'[69] package). In the face of small, potentially non-normal samples, we confirmed significant results with bootstrap confidence intervals and p-values, leading to consistent results in all but one correlation (Supplementary Fig. 10g): Here, we obtained C.I = $[−0.83, −0.02]$ and $p = 0.04$ with parametric methods, but C.I. = $[−0.85, 0.09]$ and p = 0.09 with non-parametric methods (all two-sided; note however that our directed hypothesis of an expected negative correlation supports a one-sided test with p = 0.04). Confidence intervals of the mean (Figs. 1 and 3, and Supplementary Figs. 5, 6 and 9) were calculated as 95% bootstrap confidence intervals.

**Neural network architecture and dynamics.** We simulated consecutive pairs of trials in a spiking neural network model of prefrontal cortex implemented in Brian2[70]. $N_E = 1024$ excitatory and $N_I = 256$ inhibitory leaky integrate-and-fire

neurons were connected all-to-all via synapses governed by NMDAR-, AMPAR-, and GABA$_A$R-dynamics, as described in ref. [21].

The dynamics of the membrane voltage of excitatory neurons $V_i (i = 1..N_E)$ were given by:

$$C_m \frac{dV_i}{dt} = -g_L(V_i - E_L) - g_{EE,A} \sum_j^{N_E} W_{ij}^{EE} s_j^A (V_i - E_A)$$
$$- \frac{g_{EE,N}}{1 + e^{-aV_i}/3.57} \sum_j^{N_E} W_{ij}^{EE} s_j^N (V_i - E_N) \qquad (10)$$
$$- g_{IE} \sum_j^{N_I} s_j^G (V_i - E_G) - g_{ext,E} s_{ext}(V_i - E_A) + I_i^s$$

with membrane capacitance $C_m = 0.5$ nF, leak conductance $g_L = 25$ nS, leak reversal potential $E_L = -70$ mV, AMPAR, GABA$_A$R and NMDAR reversal potentials $E_A = 0$ mV, $E_G = -70$ mV, $E_N = 0$ mV, unitary conductances $g_{ext,E} = 3.1$ nS, $g_{IE} = 2.672$ nS, $g_{EE,N} = 0.56$ nS, $g_{EE,A} = 0.502$ nS, and the NMDAR magnesium block parameter $a = 0.062$ mV$^{-1}$. In simulations of reduced NMDAR conductance, parameters $g_{EE,N}$ or respectively $g_{EI,N}$ were modulated as indicated in Fig. 3b, c, e, f and Supplementary Fig. 14.

The membrane voltage of inhibitory neurons followed:

$$C_m \frac{dV_i}{dt} = -g_L(V_i - E_L) - g_{EI,A} \sum_j^{N_E} s_j^A (V_i - E_A)$$
$$- \frac{g_{EI,N}}{1 + e^{-aV_i}/3.57} \sum_j^{N_E} s_j^N (V - E_N) \qquad (11)$$
$$- g_{II} \sum_j^{N_I} s_j^G (V_j - E_G) - g_{ext,I} s_{ext}(V_i - E_A)$$

with $C_m = 0.2$ nF, $g_L = 20$ nS, $g_{ext,I} = 2.38$ nS, $g_{II} = 2.048$ nS, $g_{EI,A} = 0.384$ nS and $g_{EI,N} = 0.424$ nS.

The kinetics of synaptic variables $s_i^A (i = 1 \cdots N_E)$, $s_i^G (i = 1 \cdots N_I)$, and $s_{ext}$ were determined by

$$\frac{ds_X}{dt} = -\frac{s_X}{\tau_X} + w \sum_i \delta(t - t_i) \qquad (12)$$

with $\tau_A = 2$ ms, $\tau_G = 10$ ms, $\tau_{ext} = 2$ ms, and the summation running over all spike times $t_i$ so that at each spike time the synaptic variable increased by a step of magnitude $w$, which was generally set to 1 except for synapses undergoing synaptic potentiation (see below). For $s_{ext}$, spike times were generated as a Poisson spike train of rate 1800 spikes s$^{-1}$ (simulating inputs from 1000 external Poisson neurons firing at 1.8 spikes s$^{-1}$ each).

The slower and saturating NMDAR synaptic variables $s_i^N (i = 1 \dots N_E)$ followed the coupled equations:

$$\frac{ds_i^N}{dt} = -\frac{s_i^N}{\tau_{N_s}} + \alpha_N x_i (1 - s_i^N) \qquad (13)$$

$$\frac{dx_i}{dt} = -\frac{x_i}{\tau_{N_x}} + w \sum_j \delta(t - t_j) \qquad (14)$$

with $\tau_{N_s} = 100$ ms, $\tau_{N_x} = 2$ ms, and $\alpha_N = 0.5$ kHz.

The strength of recurrent excitatory synapses was modulated depending on the distance in preferred location of presynaptic and postsynaptic excitatory neurons: $W_{ij}^{EE} = J(\theta_i - \theta_j)$, where $J$ is a Gaussian function (centered at $\mu = 0$ with $\sigma = 14.4$ degrees) plus a constant, tuned so that $\sum_j J(\theta_i - \theta_j) = N_E$ and $J(0) = 1.63$. As a result, neurons with similar preferred locations had 1.63 stronger weights than the average weight (Supplementary Fig. 10 for network scheme and weight profiles).

**STP rule in neural network simulations**. For connections between excitatory neurons, the spike-triggered step in AMPAR and NMDAR synaptic variables $w$ could vary individually for each specific connection: $w_{ij}$ characterized the step at the synapse from neuron $j$ onto neuron $i$. Upon synchronized pre- and post-synaptic spiking, $w_{ij}$ was slightly enhanced by an amount $\Delta_w$ that depended on the relative spike times of neuron $j$ and $i$ (Fig. 2c) to simulate an increase in probability of glutamate release[71]:

$$w_{ij} = w_{ij} + \Delta_w(t_j - t_i) \geq 1 \qquad (15)$$

The associative nature of this rule was determined by a potentiation function that required synchronization within a specific temporal window (Fig. 2d):

$$\Delta_w(t_j - t_i) = P \exp\left(-|t_j - t_i|/\tau_\Delta\right), \qquad (16)$$

with potentiation factor $P = 0.00022$ and $\tau_\Delta = 20$ ms. Changes were sustained (did not decay with time), but synapses depotentiated based on presynaptic activity[3]: at

each presynaptic spike

$$w_{ij} = w_{ij} - 0.04*(w_{ij} - 1) \qquad (17)$$

**Trial structure in neural network simulations**. We simulated 20,000 pairs of consecutive trials with independent randomized stimulus locations. Network inputs $\theta_n^s$ in trial $n$ with stimulus $s$ were slightly transformed to mimic a repulsive baseline bias away from previous stimulus locations, resulting from sensory aftereffects produced in lower-level cortical areas[29]: $\theta_n^{s'} = \theta_n^s + 1.25 \, DoG(\theta_n^d)$, where $DoG(\theta_n^d)$ is the first-derivative-of-Gaussian function with $\mu = 0$ and $\sigma = 0.8$ radians, and $\theta_n^d$ is the distance between previous and current stimulus.

Simulations started with a stimulus presentation at 0° (trial $n - 1$) for 0.25 s. After the input was removed, a delay of 1 s followed. A negative input to the whole network during 0.25 s simulated the response and removed stimulus-associated neural activity. After an ITI of 3 s, a second stimulus (trial $n$) was delivered at a random location for 0.25 s. The second delay duration was 3 s. To obtain behavioral readouts from the network, we counted each neuron's spikes during three time windows of 0.25 ms: 0–0.25 s after stimulus offset (0 s delay condition), 0.75–1 s (1 s delay), and 2.75–3 s after stimulus offset (3 s delay). The behavioral response was determined as the angular direction of the population vector of spike counts.

**Neural network behavioral analysis**. We first calculated the percentage of outlier responses and excluded outlier trials from the network's population vector responses (response error >1 radian). Circular standard deviations and serial dependence were then calculated from the network's population vector responses analogous to human error analyses. In Fig. 3a–f, bias strength was measured as the sum of bias term coefficients in the linear model

$$\theta_n^e = \beta_0 + \beta_{1,d} \text{delay}_n - \beta_2 DoG(\theta_n^d) - \beta_{3,d} \text{delay}_n DoG(\theta_n^d) + \varepsilon_n \qquad (18)$$

that fitted errors $\theta_n^e$ in trial $n$ from each parameter manipulation ($P$, $g_{EE}$, and $g_{EI}$) separately as a function of delay and $DoG(\theta_n^d)$ with $\mu = 0$ and $\sigma = 0.6$ radians.

**Hyperparameter cross-validation for neural network responses**. The value of hyperparameter $\sigma$ was determined in a cross-validation procedure for the baseline condition with $P = 0.00022$, $g_{EE} = 0.56$ nS, and $g_{EI} = 0.424$ nS, for values $\sigma \in [0.2, 1.8]$ (in radians). For each value of $\sigma$, we fitted the linear model described in Eq. (18) to 67% of trials and tested the prediction in the left-out 33% of trials. Performance for each $\sigma$ was evaluated using the mean squared error (MSE) of predictions from 1000 cross-validation repetitions. $\sigma$ was chosen to minimize the MSE of the linear model, yielding $\sigma = 0.6$ radians (Supplementary Fig. 13).

**Reporting summary**. Further information on research design is available in the Nature Research Reporting Summary linked to this article.

## Data availability

Behavioral data analyzed in this article are openly available by accessing the github repository: github.com/comptelab/serialNMDA

## Code availability

Custom code used for simulations and data analysis is openly accessible through the github repository: github.com/comptelab/serialNMDA

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

## Acknowledgements

We acknowledge support from Institute Carlos III, Spain (Ref: PIE 16/00014), CELLEX Foundation, Safra Foundation, CERCA Programme/Generalitat de Catalunya, Generalitat de Catalunya (AGAUR 2014SGR1265, 2017SGR01565), "la Caixa" Foundation (ID 100010434, under the agreement LCF/PR/HR17/52150001), and by the Spanish Ministry of Science, Competitiveness and Universities co-funded by the European Regional Development Fund (Refs: BFU 2015-65318-R, RTI2018-094190-B-I00). HS was supported by the "la Caixa" Banking Foundation (Ref: LCF/BQ/IN17/11620008), and the European Union's Horizon 2020 Marie Skłodowska-Curie grant (Ref: 713673). JB was supported by the Bial Foundation (ref: 356/19). We thank the Barcelona Supercomputing Center (BSC) for providing computing resources. This work was developed at the buildings Centro Esther Koplowitz, CELLEX, and Hospital Clinic, Barcelona. We thank Thaís Armangue, Domingo Escudero, Amaia Muñoz-Lopetegi, and Gisela Sugranyes for assistance in recruiting patients. Jaime de la Rocha, Daniel Linares for discussions, Ainhoa Hermoso-Mendizabal for comments on the manuscript, and Diego Lozano-Soldevilla for assistance during the development of the task.

## Author contributions

H.S., J.B., and A.C. designed behavioral and computational aspects of the study. J.C.F., J.D., and A.C. designed clinical aspects of the study. H.S. performed analyses of human behavior and computer simulations. H.S., J.B., and A.C. developed the computational model. H.S., A.M., L.P., and A.G.G. performed human experiments. M.R.J. and L.P. performed neuropsychological testing. J.C.F., M.R.J., H.A., and E.M.H. recruited participants for the study. H.S. and A.C. wrote the paper. H.S., J.B., J.D., and A.C. discussed the results and edited the paper. All authors reviewed the paper for intellectual content.

## Competing interests

J.D. receives royalties from Athena Diagnostics for the use of Ma2 as an autoantibody test and from Euroimmun for the use of NMDAR, GABA$_B$ receptor, GABA$_A$ receptor, DPPX and IgLON5 as autoantibody tests. All other authors declare no competing interests.
