## [Peer Review File · Nature Communications]

In this paper, Stein and colleagues use different computational models to relate effects of schizophrenia and anti-NMDAR encephalitis to serial effects in a delayed-response task. While they find no difference in actual memory performance, they do show a difference between three groups at a behavioural level: while the serial effect of probes is initially repulsive in all groups, it becomes attractive in healthy controls with increasing delay, less so in the aNMDAR group, and remains repulsive in the SCZ group. They also show that the attractive effect shows signs of improving over time with clinical recovery in the aNMDAR group. Using a network of excitatory and inhibitory neurons, they then attempt to distinguish NMDAR dysfunction affecting either synaptic plasticity, or excitatory-excitatory or excitatory-inhibitory function. They find that only the first can qualitatively capture the data.

Overall, this is an excellent, well-written and exciting paper. It combines an interesting study design with elegant computational modelling, and addresses an important question.

Issues

- My main concern relates to the impact of medication. This is very strongly correlated with the group effects, and there are trends in the CPZ equivalents in both patient groups (I imagine that pooling across the patient groups would show a significant effect). More information and data is needed to convince me that these are not pure medication effects. The current analyses and data are not sufficient in that regard. I should emphasize that the results continue to be important and relevant, but the impact of these medications need to be discussed and displayed in more detail.
- I was unsure about the importance of memory-independent repulsive biases in the E-I network. How are these important for the overall results?
- For the symptom correlations, I wonder whether the authors have considered a linear mixed effects analysis across groups with factors controlling for group.
- Please clarify if any of these data have been published before

Reviewer #2:

Remarks to the Author:

The manuscript from Stein et al. proposes a nice psychophysical and computational account of working memory deficits observed in two disorders associated with excitatory-to-inhibitory neural imbalance, i.e., anti-NMDA-R encephalitis and Schizophrenia. Based on quantitative model-fits with real data collected during a visuo-spatial delayed-response task (in the two pathological groups plus a group of healthy controls), the authors argued for the involvement of impaired NMDA-dependent candidate mechanisms, such as short-term potentiation (STP). The manuscript is well-written and I do have quite a few comments and suggestions which I hope the authors will find helpful for a revision of their paper.

- 1) Schizophrenia (SCZ) and NMDA-R encephalitis (ENC) could greatly differ in terms of underlying

biological mechanisms, and these mechanisms could also vary according to specific clinical dimensions, so why absolutely trying to find a common mechanism for both disorders (STP here) ? This should be better justified. For instance, the cognitive dimension was nicely explored here (through Working Memory deficits), but the two clinical groups seemed to have very limited psychotic symptoms for instance (Supp Table 1), a dimension which has been related to false inference in alternative frameworks (Bayesian notably) and could also be involved in the pathophysiology of the these disorders. Hardly no positive symptoms were observed for ENC patients (only negative ones). This should be discussed and acknowledged, since patients are explored during relatively stabilized (or non-acute) states.

2) ENC data seems to be explained qualitatively without disruption of the STP, even though the fit presented Figure 3 is not so good quantitatively. Could the authors clarify this point ?

3) Regarding Hyp. 1: why would the STP alteration be only on the E-E synapses ? Please clarify in the main text.

4) Regarding Hyp. 3: A decrease of g_{EE} would mean a lower E-I ratio, right ? But to my knowledge and based on the literature, this is not the case for SCZ: please clarify.

5) Regarding the task itself: Was there a preference for a particular direction in the different samples explored ? The paper seems to only compute the difference between angles, without consideration of the values of these angles.

6) In the data presented, the behavior seemed less affected in ENC participants than in SCZ participants compared with CTLs, right ? I was wondering if there was a direct experimental evidence from the literature (e.g. STP disruption of NMDA-R for ENC vs SCZ), or indirect evidence coming from behavioral studies for that pattern ?

7) Did the authors find any correlation between the scales provided Supp. Table 2 and the amount of serial bias ? Please comment and discuss.

8) Minor comments:

a. in Figure 3 g.h.i., please add the experimental data as you did in Figure 1 d.e.f.

b. Suppl. Figure 7: I think there is a mistake in the title: "n-1" instead of "n+1".

Reviewer #3:

None

Point-by-point response to the Reviewers' concerns

We thank the Reviewers for their appreciation of our work and their comments. We include a point-by-point response and several new analyses addressing these comments below. We believe that these new analyses strengthen the conclusions of our manuscript. The current revision now includes additional results based on these new analyses:

- 1) We have expanded our analysis of the impact of antipsychotic medication on patients' serial dependence. Our new approach shows that chlorpromazine equivalents (CPZ) only explain a small fraction of the observed group differences in delay-dependent serial biases and allows us to make stronger conclusions about group effects that are independent of CPZ effects. These analyses have led to a new figure (Supplementary Fig. 9) and a new paragraph in the *Results* section (*Antipsychotic medication does not explain group differences*).
- 2) We have further explored the relationship between serial bias strength and clinical symptoms of anti-NMDAR encephalitis patients, by studying it in relation with the longitudinal component. We have found correlations between psychotic symptoms and serial dependence in follow-up sessions of encephalitis patients, and a correlation of longitudinal improvement in psychotic symptoms with the normalization of biases. These new analyses are now reported in additional panels of Supplementary Fig. 10 and in the *Results* section (*Encephalitis patients' biases increase with recovery*).
- 3) We have explored potential group differences in an additional type of biases not previously reported in the manuscript (history-independent biases regarding cardinal directions). We found that all groups show similar, delay-dependent repulsion from the cardinal axes, an interesting finding that also underscores that group differences in serial dependence are not explained by general alterations in behavioral biases in this task. This control is now reported in Supplementary Fig. 11.

Moreover, our manuscript now focuses more prominently on several clinical aspects of schizophrenia and anti-NMDAR encephalitis, and how they might relate to the results of our study. To this end, we included several clarifications in *Introduction* and *Results*, and two new paragraphs in *Discussion* that lay out potential limitations to the parallels drawn between the two clinical groups of our study, alternative neurobiological explanations of our findings, and that discuss the new evidence for a relation of serial dependence with the extent of psychotic symptoms. We thank the Reviewers for their suggestions and believe that these changes in the manuscript will increase the clarity and interpretability of our work.

Reviewer #1

Remarks to the Author

In this paper, Stein and colleagues use different computational models to relate effects of schizophrenia and anti-NMDAR encephalitis to serial effects in a delayed-response task. While they find no difference in actual memory performance, they do show a difference between three groups at a behavioural level: while the serial effect of probes is initially repulsive in all groups, it becomes attractive in healthy controls with increasing delay, less so in the aNMDAR group, and remains repulsive in the SCZ group. They also show that the attractive effect shows signs of improving over time with clinical recovery in the aNMDAR group. Using a network of excitatory and inhibitory neurons, they then attempt to distinguish NMDAR dysfunction affecting either synaptic plasticity, or excitatory-excitatory or excitatory-inhibitory function. They find that only the first can qualitatively capture the data.

Overall, this is an excellent, well-written and exciting paper. It combines an interesting study design with elegant computational modelling, and addresses an important question.

We thank the Reviewer for these enthusiastic comments.

Issues

Reviewer #1, comment 1: My main concern relates to the impact of medication. This is very strongly correlated with the group effects, and there are trends in the CPZ equivalents in both patient groups (I imagine that pooling across the patient groups would show a significant effect). More information and data is needed to convince me that these are not pure medication effects. The current analyses and data are not sufficient in that regard. I should emphasize that the results continue to be important and relevant, but the impact of these medications need to be discussed and displayed in more detail.

To address the concern that medication might explain a substantial part of the observed group differences, we reasoned that group differences in serial dependence should persist after controlling for the effects that can be explained by CPZ equivalents. One control of this variable was already performed in the manuscript, where we included a multiplicative term of $CPZ \times delay \times DoG(\theta)$ in our linear model (Eq. 4.2), to then show that multiplicative and overall effects of group on serial dependence persisted consistently with the findings of Fig. 1.

However, we agree with the Reviewer that medication still explains an important amount of variance in the observed effect. With our experimental design, we cannot know whether medication causes this decrease in serial dependence, or whether both medication and

reduced serial dependence are commonly explained by the severity of the subjects' neurological or psychiatric condition. To prove that group differences are observable even in the absence of antipsychotic medication, we present now two additional analyses:

- A. We performed analyses reported in Fig. 1 (Eqs. 1-3) only on subjects who had a CPZ equivalent of 0 mg day^{-1} . This analysis included *all control* subjects ($n = 19$) and a *subgroup of encephalitis* patients ($n_{\text{enc}} = 12$ out of $n_{\text{enc}} = 16$), but *no patients with schizophrenia* (as only $n_{\text{schz}} = 1$ did not take antipsychotic medication, and the group effect cannot be estimated).
- B. We provide a conservative estimate of group effects *after* regressing out delay-dependent medication effects on serial bias in a first step:

$$\theta_n^e = \beta_0 + \beta_1 CPZ_n + \beta_2 CPZ_n delay_n - \beta_3 CPZ_n DoG(\theta_n^d) - \beta_4 CPZ_n delay_n DoG(\theta_n^d) + resid_n$$

explains all possible main and modulating effects of CPZ. Note that we do not fit random effects, as some variance between subjects will depend on the factor *group*, and we want our residuals to still contain that part of the variance.

In a second step, we then fit the mixed model described in Eq. 1 on our residuals $resid_n$:

$$\begin{aligned} resid_{nm} = & \beta_0 + \beta_1 group_{nm} + \beta_2 delay_{nm} - \beta_3 DoG(\theta_{nm}^d) \\ & + \beta_4 group_{nm} delay_{nm} - \beta_5 group_{nm} DoG(\theta_{nm}^d) - \beta_6 delay_{nm} DoG(\theta_{nm}^d) \\ & - \beta_7 group_{nm} delay_{nm} DoG(\theta_{nm}^d) \\ & + \gamma_{0m} - \gamma_{1m} delay_{nm} DoG(\theta_{nm}^d) + \varepsilon_{nm} \end{aligned}$$

Group differences obtained from this model are free of any (delay-specific) linear contributions of antipsychotic medication on serial dependence, as measured by the *CPZ equivalent*.

Below, we report results for analyses A and B.

- A. When restricting the fit of the model in Eq. 1 to the unmedicated subset of patients, the group difference of delay-dependent biases remained significant ($group \times delay \times DoG(\theta^d)$, $F(2,28) = 4.49$, $p = 0.02$ with $n_{\text{enc}} = 12$, as compared to $F(2,32) = 5.70$, $p = 0.008$ with $n_{\text{enc}} = 16$ encephalitis patients, excluding the schizophrenia group), while no significant delay-independent group differences were found in neither the subset, nor the full group of patients ($group \times DoG(\theta^d)$, $F(1,29) = 1.29$, $p = 0.27$ for $n_{\text{enc}} = 12$, compared to $F(1,33) = 3.91$, $p = 0.06$ with $n_{\text{enc}} = 16$). Delay-wise models showed that the

difference in biases between groups was strongest in 3 sec delay trials ($group \times DoG(\theta^j)$, $F(1,29) = 5.80$, $p = 0.02$ for $n_{enc} = 12$), whereas differences in shorter delays were non-significant ($group \times DoG(\theta^j)$, $F(1,30) = 0.15$, $p = 0.70$ for 0 sec delay, and $F(1,29) = 0.00$, $p = 0.99$ for 1 sec delay), confirming previously obtained results. We now include a new figure in the manuscript (Supplementary Fig. 9a-f) that illustrates this specific analysis. Suppl. Fig. 9a-f shows groupwise bias curves and random effects, as well as pairwise comparisons between individual estimates for the subgroups of individuals who did not take antipsychotic medication at the time of testing.

- B. Supplementary Fig. 9g-l shows groupwise bias curves and random effects, as well as pairwise comparisons between individual estimates for the subgroups of individuals after regressing out linear and multiplicative effects of CPZ on overall and delay-dependent serial dependence. CPZ had a significant effect on delay-independent and delay-dependent biases ($CPZ \times DoG(\theta^j)$, $F(1,52380) = 196.84$, $p < 9.0 \text{ e-}16$, and $CPZ \times delay \times DoG(\theta^j)$, $F(2,52380) = 4.97$, $p = 0.007$). Importantly however, both delay-independent and delay-dependent serial biases remained significantly reduced in both patient groups after partially regressing CPZ-dependencies from errors ($group \times DoG(\theta^j)$, $F(2,49.3) = 3.53$, $p = 0.04$, and $group \times delay \times DoG(\theta^j)$, $F(4,63.4) = 6.15$, $p = 3\text{e-}4$).

We now report these analyses in the caption of Supplementary Fig. 9, and rewrote the *Results - Antipsychotic medication does not explain group differences* paragraph to include the results in the main text.

Reviewer #1, comment 2: I was unsure about the importance of memory-independent repulsive biases in the E-I network. How are these important for the overall results?

We agree with the Reviewer that memory-independent repulsive biases are not the focus of our manuscript, but we claim that including them in our modeling provides the reader with a more comprehensive view of the behavioral results that we report. Our reasoning behind modeling repulsive baseline-biases is based on previous literature. Experimentally, the strength of attractive serial dependence can be reduced by 1) increasing inter-trial-interval length ¹, 2) shortening the current trial's working memory delay ¹ and 3) backward masking ². In all cases, the absence of attractive serial dependence reveals small repulsive biases which are commonly regarded as a result of sensory adaptation processes. This effect has been previously modeled as emerging outside (prefrontal) working memory circuits ³. Consistent with these previous findings, we observe significant repulsive biases in 0 seconds delay (Fig. 1d), for all groups. If we regard the attractive bias as an *additive*, memory-dependent effect ^{1,3}, then the absence of this effect explains why patients with schizophrenia show repulsive biases for all delays. We therefore think that our modeling decision both fits the data and provides the reader with the current understanding of the origin of repulsive biases: a memory-independent, "default" sensory bias, which, in healthy controls but not so much in patients, is overwritten by attractive working-memory biases.

We now include an additional sentence in the *Results - NMDAR hypofunction in a prefrontal working memory circuit* section of our paper to clarify this reasoning, and add two additional references: “*To mimic memory-independent repulsive biases^{1,2}, current stimulus inputs were slightly shifted away from previous stimulus values by a fixed value³ (Methods). This shift represents adaptation effects in sensory regions and is therefore not affected by local circuit alterations in prefrontal cortex.*”

Reviewer #1, comment 3: For the symptom correlations, I wonder whether the authors have considered a linear mixed effects analysis across groups with factors controlling for group.

We thank the Reviewer for their suggestion. Indeed, as already pointed out in the first issue raised by Reviewer #1 (in the context of medication), our analyses performed for each group separately could be underpowered. As per the Reviewer’s suggestion, we instead fit linear models for each clinical measure x_m for subject m , pooling subjects across groups and controlling for group:

$$bias_m = \beta_0 + \beta_1 group_m + \beta_2 x_m + \varepsilon_m.$$

Depending on the clinical measure, we pooled subjects in two different ways:

- A. For psychosis-related measures x_m (i.e. CPZ equivalents and PANSS positive, negative and general scales), in which variability was consequently equal or close to zero in healthy controls, we fitted models on *patients’ biases only* (combining $n = 16$ enc, and $n = 17$ schz)
- B. For all other measures (GAF, YMRS and HAM-D), we fitted models on *all subjects’ biases* (combining $n = 19$ ctrl, $n = 16$ enc, and $n = 17$ schz).

None of the clinical scales showed a significant relation with bias strength; however, as correctly predicted by the Reviewer in comment 1, CPZ equivalents significantly correlated with biases (CPZ, $F(1,30) = 6.52$, $p = 0.02$). We address this result and the controls performed for comment 1 in the new *Results - Antipsychotic medication does not explain group differences* paragraph, and report the analyses obtained from these models in the caption of Supplementary Fig. 8.

Reviewer #1, comment 4: Please clarify if any of these data have been published before

Behavioral data of 14 control participants has been reported in⁴. We indicate this circumstance in the statistics checklist and in the *Methods* section (*Experimental Procedures - Sample*): “*Behavioral data from $n=14$ healthy controls has been reported previously⁴*”

Finally, we want to point out to the Reviewer that small numerical changes in our analyses resulted from a correction in the preprocessing of our dataset, leading to the inclusion of several more trials (~ 10 trials). As the Reviewer can see in the track changes, this change affects our statistics minimally (on the order of 10^{-2} for reported F- and t-statistics) and have no implications for any of the results reported in the earlier submission.

Reviewer #2

Remarks to the Author

The manuscript from Stein et al. proposes a nice psychophysical and computational account of working memory deficits observed in two disorders associated with excitatory-to-inhibitory neural imbalance, i.e., anti-NMDA-R encephalitis and Schizophrenia. Based on quantitative model-fits with real data collected during a visuo-spatial delayed-response task (in the two pathological groups plus a group of healthy controls), the authors argued for the involvement of impaired NMDA-dependent candidate mechanisms, such as short-term potentiation (STP). The manuscript is well-written and I do have quite a few comments and suggestions which I hope the authors will find helpful for a revision of their paper.

We thank the reviewer for their appreciation of our work, and for the constructive comments that have introduced important improvements in our manuscript.

Issues

Reviewer #2, comment 1: Schizophrenia (SCZ) and NMDA-R encephalitis (ENC) could greatly differ in terms of underlying biological mechanisms, and these mechanisms could also vary according to specific clinical dimensions, so why absolutely trying to find a common mechanism for both disorders (STP here)? This should be better justified. For instance, the cognitive dimension was nicely explored here (through Working Memory deficits), but the two clinical groups seemed to have very limited psychotic symptoms for instance (Supp Table 1), a dimension which has been related to false inference in alternative frameworks (Bayesian notably) and could also be involved in the pathophysiology of these disorders. Hardly no positive symptoms were observed for ENC patients (only negative ones). This should be discussed and acknowledged, since patients are explored during relatively stabilized (or non-acute) states.

The Reviewer points us to several sources of heterogeneity between the patient groups that may question our integrated interpretation on the basis of one common mechanistic alteration:

- A. Biological mechanisms underlying anti-NMDAR encephalitis and schizophrenia could greatly differ.
- B. The biological mechanisms that underlie reduced serial dependence could differ between patient groups.
- C. One example of patient heterogeneity in our data that could be related to different biological mechanisms is the limited scope of positive symptoms (as measured by the PANSS positive scale) in encephalitis patients, compared to patients with schizophrenia. This could be of importance as positive symptoms are positively correlated with an overweighting of sensory evidence compared to prior information in inference tasks, and is explained by disrupted inhibition in bottom-up cortical pathways in Bayesian modeling frameworks.

We acknowledge the Reviewer's careful assessment of our data and modeling. Below, we address the concerns issued above:

- A. As pointed out by the reviewer, anti-NMDAR encephalitis and schizophrenia are two different diseases, with different etiology⁵⁻⁷. We now point this out explicitly in the Discussion (*"However, substantial neurobiological heterogeneity must underlie the differences in epidemiology and longitudinal development of schizophrenia and autoimmune anti-NMDAR encephalitis^{5"}*). However, the two diseases present with such a similar set of symptoms (notably psychosis and neuropsychological symptoms affecting executive functions and memory⁸) that they are often initially misdiagnosed^{9,10}. This convergence of clinical and neuropsychological symptoms suggests that among the distinct cascade of mechanistic alterations of the two diseases, there could be a common substrate in what concerns cognitive and/or perceptual processing. This common substrate is likely to involve the NMDA receptor, based on the long-standing hypothesis of NMDAR hypofunction in schizophrenia¹¹. Thus, the comparison of these two diseases is not an arbitrary choice for this study, but a sustained theme in the ever-expanding literature on anti-NMDAR encephalitis and schizophrenia^{5,6,10,12,13}. We realize that we did not emphasize this line of argumentation in our Introduction, and we now sharpened it in our revision, both in the *Introduction* (*"The prevalence of positive symptoms during early stages of the disease causes frequent misdiagnosis as a schizophrenia spectrum disorder^{9,10}."*) and *"Due to the parallels in neurobiology, clinical aspects and cognition of the two diseases, we expected..."*) and in *Discussion* (*"In this study, we assessed working memory alterations in two patient groups linked to NMDAR hypofunction, and hypothesized that their shared clinical and neurobiological features should be reflected in qualitatively similar behavioral patterns."*).
- B. Based on the evidence presented in A., our hypothesis was that in exploring memory function in these two patient populations side by side, we would expect similar effects in both groups. Indeed, we found that serial dependence was affected in the same

direction in the two groups (reduced attractive serial dependence). The most parsimonious explanation is that this effect relies on a common NMDAR-dependent mechanism, as we were able to show in a computational model. However, as the Reviewer points out, we cannot rule out that similar serial dependence alterations could be caused by different mechanisms in the two populations, for instance reduced NMDAR-dependent transmission in E-E connections in encephalitis (see comment 2 below) and reduced NMDAR-dependent STP in schizophrenia. We now declare this possibility in the *Results* section (“*While this manipulation can qualitatively reproduce decreased delay-dependent biases in the encephalitis group, ...*”) and in *Discussion* (“*Under this reasoning, we cannot exclude that distinct biological mechanisms in our two patient groups might lead to convergent patterns of working memory processing. For instance, our modeling shows that encephalitis patients’ biases could also be explained qualitatively by a reduced excitation-to-inhibition ratio in the memory circuit...*”).

- C. The Reviewer is correct in that our two patient populations differ in the PANSS positive symptoms scale. However, we do not think that this by itself indicates a fundamental mechanistic difference. As mentioned in A, anti-NMDAR encephalitis presents initially with strong psychotic symptoms, which disappear upon successful treatment of the acute condition. Positive symptoms per se do not distinguish the diseases, but instead a more clear distinction is their different time course: only rarely do anti-NMDAR encephalitis patients show recurrent outbreaks with acute psychosis. To have the collaboration of our participants in this long working memory task, both patient populations were explored in a stabilized phase of the disease where their positive symptoms were low, and we took advantage of the gradual recovery of anti-NMDAR encephalitis patients to validate the evolution to normality of serial dependence alterations. We note that based on the new analyses that we provide in this revision (see Comment 7 below), there is a correlation between serial dependence and positive symptoms in the follow-up session of anti-NMDAR encephalitis patients, consistent with a relationship between NMDARs, STP, serial dependence, and mild positive symptoms. Along this line, the group differences in positive symptoms go in the same direction as the group differences in serial dependence. Both of these points are now included in the Discussion: “*First, as anti-NMDAR encephalitis patients recovered, their biases normalized in the direction of healthy controls. Second, the amount of this normalization correlated across patients with their improvement on a scale that measures positive symptoms, indicating a potential relation between psychotic symptoms and reductions in serial dependence. Third, both the alterations in serial dependence and the strength of positive symptoms were higher for patients with schizophrenia than for the anti-NMDAR encephalitis group.*” However, the possibility still remains that microcircuit alterations manifest differently in more acute stages (with more psychotic symptoms) of the diseases, as the Reviewer suggests. For instance, acute NMDAR hypofunction could affect both STP and EI balance and generate more positive symptoms, consistent with the results of Bayesian models^{14,15}, while non-acute

or milder NMDAR hypofunction in more stabilized stages of the disease could primarily affect STP without affecting much the EI balance (possibly as a long-lasting, residual reduction in STP that occurs following initial imbalances in E-I ratio), resulting in altered serial dependence. This possibility is explained in the Discussion: *“Finally, pharmacological studies would clarify if the alterations in serial dependence occur as a result of acute NMDAR hypofunction or whether they depend on compensatory changes in STP that arise after early, acute phases of cortical excitation/inhibition imbalance in these diseases (e.g., as a long-term adjustment of the probability of presynaptic neurotransmitter release).”*

Reviewer #2, comment 2: ENC data seems to be explained qualitatively without disruption of the STP, even though the fit presented in Figure 3 is not so good quantitatively. Could the authors clarify this point?

We acknowledge the Reviewer’s concern. As mentioned in comment 1, we now include a *Discussion* paragraph about potential differences between encephalitis and schizophrenia patients from a clinical perspective, and our modeling results in light of such potential heterogeneity between patient groups: *“(…) Under this reasoning, we cannot exclude that distinct biological mechanisms in our two patient groups might lead to convergent patterns of working memory processing”* In particular, we mention the Reviewer’s concern explicitly in both *Results - Reduced STP but not altered E-I balance disrupts memory biases* (*“While this manipulation can qualitatively reproduce decreased delay-dependent biases in the encephalitis group, …”*) and *Discussion* (*“For instance, our modeling shows that encephalitis patients’ biases could also be explained qualitatively by a reduced excitation-to-inhibition ratio in the memory circuit…”*).

Reviewer #2, comment 3: Regarding Hyp. 1: why would the STP alteration be only on the E-E synapses ? Please clarify in the main text.

Although studies in hippocampus have shown evidence for NMDAR-dependent LTP in inhibitory interneurons¹⁶, the phenomenon of NMDAR-dependent STP that we implement in our simulations^{17,18} to our knowledge has not been observed in interneurons. This is the reason why we are choosing to model this mechanism only on E-E synapses in our main figure (Fig. 3). The issue raised by the Reviewer is however well taken, and lack of evidence does not mean we cannot evaluate in our modeling the possible impact of this possible mechanism. We therefore decided to reproduce our results shown in Fig. 3 in a model in which STP also strengthens excitatory-to-inhibitory connections during working memory delay. The dynamics of E-I weights underlie the same equations and parameters as in E-E weights. We observed that STP at inhibitory neurons led to a disinhibition of the circuit (potentially through a fast increase in recurrent inhibitory activity), which is why we decreased the g_{EE} parameter to avoid the network to become unstable. Results of these simulations are now reported in *Supplementary Fig. 15*; they demonstrate that our findings from main Fig. 3 generalize to

networks with STP (and its disruption through NMDAR-dysfunction) in pyramidal and inhibitory interneurons.

Reviewer #2, comment 4: Regarding Hyp. 3: A decrease of g_{EE} would mean a lower E-I ratio, right? But to my knowledge and based on the literature, this is not the case for SCZ: please clarify.

We agree with the Reviewer that most studies found in the current literature point to an increased E-I ratio in patients with schizophrenia¹⁹⁻²¹, and this justifies exploring the role of g_{EI} in our model. However, there is also significant evidence from congruent findings in human patient and pharmacology studies, and primate pharmacology, of decreased prefrontal pyramidal cell activity, thus a decreased E-I ratio, during task performance. Driesen and colleagues (2008)²² report decreased prefrontal BOLD (spatial) working memory delay activity in patients with schizophrenia, consistent with reduced recurrent excitation. Moreover, this result is replicated in healthy controls who perform a working memory task under the influence of ketamine²³. Similar results are reported in a meta-analysis of n-back working memory fMRI activity²⁴ that also points to a potential exclusivity of reduced delay-activity in PFC (as compared to other regions, such as anterior cingulate). Consistent with this, monkey studies find reduced firing in delay-active PFC cells after both local and systemic blockade of NMDAR (while spontaneous, non-task-related firing increased or stayed at the same level)²⁵. In addition, this hypothesis receives support from human post-mortem assessment of dendritic spine density in pyramidal PFC layer 3 neurons^{26,27}, which showed reduced numbers of spines in patients with schizophrenia, suggesting deficits in recurrent excitatory circuitry in PFC. These findings underscore the importance of assessing signatures of altered E-I ratio in the respective cognitive or behavioral context, and the relevance of differentiating global and local circuit alterations.

We now mention these findings more explicitly and cite^{23,25} in the main text (*Discussion*) to strengthen the reasoning behind hypothesis III (reduced g_{EE}): *“For instance, our modeling shows that encephalitis patients’ biases could also be explained qualitatively by a reduced excitation-to-inhibition ratio in the memory circuit (Fig. 3f), consistent with task-related fMRI BOLD activity in ketamine²³, and the effect of NMDAR antagonists on single-cell firing rates in monkey PFC²⁵”*.

Reviewer #2, comment 5: Regarding the task itself: Was there a preference for a particular direction in the different samples explored? The paper seems to only compute the difference between angles, without consideration of the values of these angles.

We agree that in our manuscript, the dimension of overall angular preferences remains relatively unexplored. We did not consider this analysis, given that working memory accuracy is high in all groups and outlier values are very low. To explore potential individual and group differences in preferred angular positions, we performed two additional analyses:

- A. To explore each subject's delay-resolved preference for one specific angular direction, we analyzed the vector mean of all responses given in each delay (without calculating the difference to the target position, and without excluding outlier trials). The vector's direction indicates each subject's preferred angular position in each delay, and the length of these vectors shows the strength of this preference. To explore group differences in how pronounced angular preferences were, we first corrected for anisotropies in the stimulus sampling distribution for each delay. For this, we measured the magnitude of the vector mean of responses, v_r , and the magnitude of the vector mean of stimuli, v_s , for each subject and delay, and regressed (delay-specific) v_s from v_r :

$$v_r = \beta_0 + \beta_1 v_s + \beta_2 \text{delay} + \beta_3 v_s \text{delay} + \text{resid}$$

Then, we performed an ANOVA on residual vector strength *resid* for each subject and delay:

$$\text{resid} = \beta_0 + \beta_1 \text{group} + \beta_2 \text{delay} + \beta_3 \text{group} * \text{delay} + \epsilon$$

- B. It has been reported (e.g. ²⁸⁻³⁰) that responses in delayed-response tasks with continuous response-dimensions show systematic biases with respect to the cardinal directions (attraction/repulsion from locations 0°, 90°, 180°, 270°). To assess these effects in our data, we binned response errors by target position and calculated the mean error for each bin (per subject and delay). Then, we ran an ANOVA on the s.d. of the resulting subject- and delay-wise curves to assess if the strength of cardinal biases differs between groups.

Below, we report results of analyses A and B. None of these analyses show group differences or provide alternative explanations to the results reported in the manuscript.

- A. Fig. R1 shows mean response vectors for each group and delay. Visually, we see no strong preferences for a particular angle in subject's responses, with magnitudes of $v_r = 0.07 \pm 0.03$ (ctrl), $v_r = 0.05 \pm 0.03$ (enc), and $v_r = 0.07 \pm 0.03$ (schz, all mean \pm std) ($v_r = 1$ denotes that a subject always reported the same angular location in a given delay), a fact that is also captured by the subjects' high precision (and small number of outliers) in our task. Much of these response anisotropies were explained by inhomogeneities in stimulus distribution, which arise from the random generation of stimuli with relatively small samples for each delay: An ANOVA showed that stimulus anisotropy strongly explained response anisotropy (v_s , $F(1,150) = 412.22$, $p = 7e-45$). Residual anisotropy of responses did not differ between groups (*group*, $F(2,147) = 1.55$, $p = 0.22$) or as an interaction of group and delay (*group* \times *delay*, $F(4,147) = 0.82$, $p = 0.52$).
- B. In Supplementary Fig. 11, we show group- and delay-wise averages and standard errors of response errors by angular stimulus position. Our analysis reveals interesting repulsive effects from cardinal positions that increased with delays, and reached a

strength of up to 10°. This effect, however, did not differ between groups. We compared subject- and delay-wise standard deviations of the curves shown in Supplementary Fig. 11 to assess the strength of repulsion from cardinal directions statistically. An ANOVA showed highly significant delay effects ($F(2,147) = 72.45, p < 1e-16$), but no overall group differences ($F(2,147) = 1.72, p = 0.18$) or delay-dependent group differences ($F(4,147) = 0.16, p = 0.96$). We include this in the Supplementary Material as a control analysis that underscores the specificity of the behavioral alteration reported in our manuscript.

Fig R1 | Angular preferences in subjects' responses, by group and delay

Reviewer #2, comment 6: In the data presented, the behavior seemed less affected in ENC participants than in SCZ participants compared with CTLs, right? I was wondering if there was a direct experimental evidence from the literature (e.g. STP disruption of NMDA-R for ENC vs SCZ), or indirect evidence coming from behavioral studies for that pattern?

To our knowledge, our study is the first to directly compare behavioral or psychiatric patterns in patients with schizophrenia to those in anti-NMDAR encephalitis patients. Quantitative

comparisons between psychiatric or neuropsychological scores in patient groups of different studies are therefore problematic, as the groups will not be matched and the methodology between studies can differ widely. Differences between the two patient groups in our sample are systematically showing more severe symptoms in the SCZ group (Supplementary Table 1), in line with our observation of graded alterations in serial biases.

As for gradual differences in neurobiological deficits for these two diseases, there is no study in the literature addressing differences of these diseases in post-mortem in-vitro experiments. Also, there are no comparative studies in animal models, in particular for the lack of an accurate model for schizophrenia, and the relatively recent development of an animal model for anti-NMDAR encephalitis. While it has been shown that LTP is reduced in mice that are infused with patients' NMDAR antibodies ³¹, there are currently no in-vitro studies of STP in these animal models. Again, even if these studies existed, quantitative statements about STP would be hard to make based on comparing qualitatively defined animal models. Therefore, we cannot explain this aspect of our results with direct evidence from the literature.

A potential explanation comes from the longitudinal assessment of encephalitis patients' biases: Serial dependence seems to evolve longitudinally in patients, towards more attractive biases as recovery progresses. Assuming a monotonic positive trend, patients could start from more repulsive biases (similar to schizophrenia) in more acute stages of anti-NMDAR encephalitis, and reach control participants' level when fully recovered. From this point of view, the difference that we observe between encephalitis and schizophrenia patients would not reflect a fundamental difference between the two diseases, but the result of the recovery of mechanisms in anti-NMDAR encephalitis towards the control condition.

Reviewer #2, comment 7: Did the authors find any correlation between the scales provided Supp. Table 2 and the amount of serial bias? Please comment and discuss.

We thank the Reviewer for suggesting this analysis. Indeed, we had not analyzed correlations of clinical scales with single-subject bias estimates for the follow-up session. The smaller sample in the reduced follow-up dataset precluded a random effects analysis, as done for all other analyses in the manuscript. We now estimate single-subject memory-dependent biases by fitting $DoG(\theta)$ directly to subject-, session- and delay-wise data (only for 3 sec-delay trials). We then correlate encephalitis patients' bias strength with clinical scales (parallel to the analysis presented in Supplementary Fig. 8). Interestingly, in this analysis, positive symptoms measured on the PANSS scale correlated negatively with bias strength for the follow-up session in encephalitis patients (but not for the more acute baseline session), although it has to be kept in mind that this correlation is based on very low PANSS pos scales with little variability (compare Supplementary Table 2). We now report this finding in *Results - Encephalitis patients' biases increase with recovery* (“*Interestingly, for this subsample of encephalitis patients, positive and general symptoms measured in the PANSS scale correlated with serial dependence in the follow-up session...*”) and discuss it critically in the *Discussion* (“*We found several indicators of clinical relevance for our finding...*” and “*Still, studies with*

larger sample sizes are needed to confirm the relation of psychotic symptoms and reduced serial biases at the subject-level, which in our study did not reach significance for two out of three analyses in patients with schizophrenia and anti-NMDAR encephalitis”).

The Reviewer’s comment pointed us to another interesting question: Maybe coarse measures and within-group heterogeneity of biases and clinical scales could partially explain our failure to find meaningful relations between them. In contrast, a within-subject comparison might be more sensitive to meaningful improvements in these measures. Therefore, we designed a new analysis to correlate change scores in clinical scales with how strongly each subject’s 3 sec-delay bias estimate would increase for the follow-up session. This analysis is now presented in Supplementary Fig. 10 g-l. It shows that in fact, subjects who showed *more improvement* of positive symptoms (measured by a decrease in PANSS pos scores) between baseline and follow-up session tended to also have a *stronger increase* in 3 sec-delay serial dependence estimates for the follow-up session, as reflected by the negative correlation between “ Δ bias” and “ Δ PANSS pos”. We now report this finding in *Results - Encephalitis patients’ biases increase with recovery* (“*Moreover, patients with a stronger longitudinal normalization of biases improved more on the scale of positive symptoms (PANSS pos) in the follow-up session, when compared to the baseline session.*”) and discuss it (*Discussion*) (“*We found several indicators of clinical relevance for our finding...*” and “*(...) Second, the amount of this normalization correlated across patients with their improvement on a scale that measures positive symptoms, indicating a potential relation between psychotic symptoms and reductions in serial dependence*”).

Minor comments

Reviewer #2, comment 8: in Figure 3 g.h.i., please add the experimental data as you did in Figure 1 d.e.f.

We thank the Reviewer for their suggestion. We now removed error shading for simulation bias curves in Fig. 3g-i and added bias curves estimated from participants’ behavior.

Reviewer #2, comment 9: Suppl. Figure 7: I think there is a mistake in the title: “n-1” instead of “n+1”.

To clarify this comment of the Reviewer, we would like to point out that the goals of the analyses presented in Supplementary Fig. 7 were two-fold: In subplots **a-c**, we address the question of how far serial dependence reaches back in trial history, and whether there were group differences in this time scale. In contrast, in subplots **d-f**, we investigated whether serial dependence was present when relating current to *future* stimulus positions in trial $n+1$. This analysis of course does not measure the dependence of current on previously held memories, but is designed to detect potential spurious trial-to-trial correlations in responses (as proposed

by ³²). If there was a significant bias to future stimuli in trial $n+1$, this would indicate that serial dependence to stimuli $n-1$ might be confounded by common response correlations underlying both effects of stimuli in trials $n-1$ and $n+1$. To make this logic more explicit, we now write: “We investigated whether serial dependence to stimulus $n-1$ and group differences in biases could be explained by general response correlations. *To detect potential spurious correlations across trials, we replaced previous-current distances (between trial n and trial $n-1$) in Eq. 1 with future-current distances (between trial n and trial $n+1$), as proposed in ³².*”

Finally, we want to point out to the Reviewer that small numerical changes in our analyses resulted from a correction in the preprocessing of our dataset, leading to the inclusion of several more trials (~ 10 trials). As the Reviewer can see in the track changes, this change affects our statistics minimally (on the order of 10^{-2} for reported F- and t-statistics) and have no implications for any of the results reported in the earlier submission.

References

1. Bliss, D. P., Sun, J. J. & D'Esposito, M. Serial dependence is absent at the time of perception but increases in visual working memory. *Sci. Rep.* **7**, 14739 (2017).
2. Fornaciai, M. & Park, J. Spontaneous repulsive adaptation in the absence of attractive serial dependence. *J. Vis.* **19**, 21 (2019).
3. Bliss, D. P. & D'Esposito, M. Synaptic augmentation in a cortical circuit model reproduces serial dependence in visual working memory. *PLoS ONE* **12**, e0188927 (2017).
4. Barbosa, J. et al. Interplay between persistent activity and activity-silent dynamics in prefrontal cortex during working memory. *BioRxiv* (2019). doi:10.1101/763938
5. Masdeu, J. C., Dalmau, J. & Berman, K. F. NMDA receptor internalization by autoantibodies: A reversible mechanism underlying psychosis? *Trends Neurosci.* **39**, 300–310 (2016).
6. Kayser, M. S. & Dalmau, J. Anti-NMDA receptor encephalitis, autoimmunity, and psychosis. *Schizophr. Res.* **176**, 36–40 (2014).
7. Oviedo-Salcedo, T. et al. Absence of cerebrospinal fluid antineuronal antibodies in schizophrenia spectrum disorders. *Br. J. Psychiatry* **212**, 318–320 (2018).
8. Finke, C. et al. Cognitive deficits following anti-NMDA receptor encephalitis. *J. Neurol. Neurosurg. Psychiatr.* **83**, 195–198 (2012).
9. Steiner, J. et al. Increased prevalence of diverse N-methyl-D-aspartate glutamate receptor antibodies in patients with an initial diagnosis of schizophrenia: specific relevance of IgG NR1a antibodies for distinction from N-methyl-D-aspartate glutamate receptor encephalitis. *JAMA Psychiatry* **70**, 271–278 (2013).
10. Maneta, E. & Garcia, G. Psychiatric manifestations of anti-NMDA receptor encephalitis: neurobiological underpinnings and differential diagnostic implications. *Psychosomatics* **55**, 37–44 (2014).
11. Olney, J. W., Newcomer, J. W. & Farber, N. B. NMDA receptor hypofunction model of schizophrenia. *J. Psychiatr. Res.* **33**, 523–533 (1999).
12. Weickert, C. S. & Weickert, T. W. What's hot in schizophrenia research? *Psychiatr. Clin. North Am.* **39**, 343–351 (2016).
13. Lennox, B. R., Coles, A. J. & Vincent, A. Antibody-mediated encephalitis: a treatable cause of schizophrenia. *Br. J. Psychiatry* **200**, 92–94 (2012).
14. Jardri, R. & Denève, S. Circular inferences in schizophrenia. *Brain* **136**, 3227–3241 (2013).
15. Jardri, R. et al. Are hallucinations due to an imbalance between excitatory and inhibitory influences on the brain? *Schizophr. Bull.* **42**, 1124–1134 (2016).
16. Kullmann, D. M. & Lamsa, K. P. Long-term synaptic plasticity in hippocampal interneurons. *Nat. Rev. Neurosci.* **8**, 687–699 (2007).
17. Volianskis, A. et al. Different NMDA receptor subtypes mediate induction of long-term potentiation and two forms of short-term potentiation at CA1 synapses in rat hippocampus in vitro. *J Physiol (Lond)* **591**, 955–972 (2013).
18. Erickson, M. A., Maramba, L. A. & Lisman, J. A single brief burst induces GluR1-dependent associative short-term potentiation: a potential mechanism for

- short-term memory. *J. Cogn. Neurosci.* **22**, 2530–2540 (2010).
19. Grent-'t-Jong, T. et al. Resting-state gamma-band power alterations in schizophrenia reveal E/I-balance abnormalities across illness-stages. *elife* **7**, (2018).
 20. Uhlhaas, P. J. & Singer, W. Neuronal dynamics and neuropsychiatric disorders: toward a translational paradigm for dysfunctional large-scale networks. *Neuron* **75**, 963–980 (2012).
 21. Lisman, J. Excitation, inhibition, local oscillations, or large-scale loops: what causes the symptoms of schizophrenia? *Curr. Opin. Neurobiol.* **22**, 537–544 (2012).
 22. Driesen, N. R. et al. Impairment of working memory maintenance and response in schizophrenia: functional magnetic resonance imaging evidence. *Biol. Psychiatry* **64**, 1026–1034 (2008).
 23. Driesen, N. R. et al. The impact of NMDA receptor blockade on human working memory-related prefrontal function and connectivity. *Neuropsychopharmacology* **38**, 2613–2622 (2013).
 24. Glahn, D. C. et al. Beyond hypofrontality: a quantitative meta-analysis of functional neuroimaging studies of working memory in schizophrenia. *Hum. Brain Mapp.* **25**, 60–69 (2005).
 25. Wang, M. et al. NMDA receptors subserve persistent neuronal firing during working memory in dorsolateral prefrontal cortex. *Neuron* **77**, 736–749 (2013).
 26. Konopaske, G. T., Lange, N., Coyle, J. T. & Benes, F. M. Prefrontal cortical dendritic spine pathology in schizophrenia and bipolar disorder. *JAMA Psychiatry* **71**, 1323–1331 (2014).
 27. Glantz, L. A. & Lewis, D. A. Decreased dendritic spine density on prefrontal cortical pyramidal neurons in schizophrenia. *Arch. Gen. Psychiatry* **57**, 65–73 (2000).
 28. Shin, H., Zou, Q. & Ma, W. J. The effects of delay duration on visual working memory for orientation. *J. Vis.* **17**, 10 (2017).
 29. Wei, X.-X. & Stocker, A. A. A Bayesian observer model constrained by efficient coding can explain “anti-Bayesian” percepts. *Nat. Neurosci.* **18**, 1509–1517 (2015).
 30. Lipinski, J., Simmering, V. R., Johnson, J. S. & Spencer, J. P. The role of experience in location estimation: Target distributions shift location memory biases. *Cognition* **115**, 147–153 (2010).
 31. Planagumà, J. et al. Human N-methyl D-aspartate receptor antibodies alter memory and behaviour in mice. *Brain* **138**, 94–109 (2015).
 32. Cicchini, G. M., Anobile, G. & Burr, D. C. Compressive mapping of number to space reflects dynamic encoding mechanisms, not static logarithmic transform. *Proc Natl Acad Sci USA* **111**, 7867–7872 (2014).

Reviewers' Comments:

Reviewer #2:

Remarks to the Author:

I very much enjoyed reading this revised version of the work as well as your replies. Hypotheses 1 and 3 have been clarified, and new analyses on angular preference and bias changes with recovery in NMDAR encephalitis patients really strengthen the findings. I have no further comment or request.

Point-by-point response to the Reviewers' concerns

Reviewer #2

Remarks to the Author

I very much enjoyed reading this revised version of the work as well as your replies. Hypotheses 1 and 3 have been clarified, and new analyses on angular preference and bias changes with recovery in NMDAR encephalitis patients really strengthen the findings. I have no further comment or request.

We thank the Reviewer for their positive feedback, and are happy to have responded satisfactorily to their previous comments.